# Low circulating miR-190a-5p predicts progression of chronic kidney disease

David P. Baird [1,7], Jinnan Zang[2,7], Katie L. Connor[3,7], Oliver Teenan[3], Maximilian Reck [3], Carolynn Cairns[3], Callum Sutherland[3], Rachel M. B. Bell[3], Jamie P. Traynor [4], Ryan Wong[5], David A. Ferenbach [1], Jeremy Hughes[1], Patrick B. Mark [4,6], Alexander P. Maxwell [2], Gareth J. McKay[2], David A. Simpson [5,8], Bryan R. Conway [3,8] & Laura Denby [3,8] ✉

MicroRNAs may act as diagnostic and prognostic biomarkers of chronic kidney disease and are functionally important in disease pathogenesis. To identify novel microRNA biomarkers, we performed small RNA-sequencing on plasma from individuals with type 2 diabetes, with and without chronic kidney disease. MiR-190a-5p abundance was significantly lower in the circulation of type 2 diabetic patients with reduced function compared to those with normal kidney function. In an independent cohort of patients with chronic kidney disease of diverse aetiology, miR-190a-5p abundance predicted disease progression in individuals with no or moderate albuminuria (< 300 mg/mmol). miR-190a-5p expression in kidney biopsy tissue correlated with the level of miR-190a-5p in the circulation and with estimated glomerular filtration rate, tubular mass and negatively with histological fibrosis. Administration of a miR-190a-5p mimic in a murine ischaemia-reperfusion injury model in male mice reduced tubular injury and fibrosis and increased expression of genes associated with tubular health. Our analyses suggest that miR-190a-5p is a biomarker of tubular cell health, low circulating levels may predict chronic kidney disease progression independent of existing risk factors and strategies to preserve miR-190a-5p may be an effective treatment for restoring tubular cell health following kidney injury.

Chronic Kidney Disease (CKD) affects 1 in 10 people worldwide, with prevalence continuing to rise in part due to the increasing incidence of diabetes[1]. The number of patients with diabetes has more than doubled since 1998[2], with almost 90% of those diagnosed having type 2 diabetes mellitus (T2DM)[3]. In the UK, more than 30% of incident end-stage kidney disease (ESKD)(need for dialysis or kidney transplantation) is due to diabetic kidney disease (DKD)[4], while this figure is almost 50% in the US[5].

Measurements of serum creatinine-based estimated glomerular filtration rate (eGFR) and urine albumin to creatinine ratio (ACR) represent the current best practice for evaluating CKD and assessing prognosis[6,7]. However, the pathophysiology of CKD is multifactorial, and many patients with CKD have no or low levels of albuminuria, even in kidney diseases characterised by albuminuria, such as diabetic nephropathy[8]. Furthermore, progressive decline in kidney function

[1]Centre for Inflammation Research, Institute for Regeneration and Repair, University of Edinburgh, Edinburgh, United Kingdom. [2]Centre for Public Health, Queen's University Belfast, Royal Victoria Hospital, Belfast, United Kingdom. [3]Centre for Cardiovascular Science, Edinburgh Kidney, Institute for Neuroscience and Cardiovascular Research, University of Edinburgh, Edinburgh, United Kingdom. [4]Glasgow Kidney and Transplant Unit, Queen Elizabeth University Hospital, Glasgow, United Kingdom. [5]Wellcome-Wolfson Institute for Experimental Medicine, Queen's University Belfast, Belfast, United Kingdom. [6]School of Cardiovascular and Metabolic Health, University of Glasgow, Glasgow, United Kingdom. [7]These authors contributed equally: David P. Baird, Jinnan Zang, Katie L. Connor. [8]These authors jointly supervised this work: David A. Simpson, Bryan R. Conway, Laura Denby. ✉e-mail: Laura.Denby@ed.ac.uk

may be observed even in those with normoalbuminuria[9]. Hence, it is imperative to identify novel biomarkers that can add value to existing clinical and biochemical risk predictors.

MicroRNAs (miRs) play important regulatory roles in diverse biological processes including metabolism, autophagy, inflammation and fibrosis[10–12]. They have been implicated in the pathogenesis of kidney disease[13–16] and may also represent therapeutic targets[17–19]. Due to their stability in body fluids, miRs have been widely assessed as potential biomarkers[20]. Quantification of miRs in the urine and circulation[21–23] can predict kidney outcomes in patients with CKD due to immunoglobulin A nephropathy (IgAN)[24], T2DM[25,26] and predict CKD onset in those with normal kidney function[27]. However, such studies typically employ individual miRs selected according to known biology, which may not represent the optimal biomarker. The development of high-throughput next-generation sequencing (NGS) technologies offers new opportunities for unbiased quantification of miRs[28,29] and has led to the identification of potential miR-based biomarkers, including for paediatric transplant rejection, CKD and IgAN[30–32].

The aims of this study were to characterise the circulating miR-Nome using an unbiased measurement of global miR expression profiles in individuals with T2DM with or without kidney disease. We further sought to assess their potential as prognostic biomarkers in a cohort of patients with unselected CKD. Finally, we employed animal and in vitro models to characterise the pathophysiological role of the most promising candidate, miR-190a-5p.

## Results

### Identification of miR-190a-5p as a candidate biomarker of CKD

To identify miRs that may represent novel CKD biomarkers in patients with diabetes, we performed small RNA-sequencing on the plasma of patients who had type 2 diabetes with CKD (T2DKD, $n = 9$) and without CKD (T2DNRF, $n = 13$) and non-diabetic controls with normal kidney function (NDNRF, $n = 11$) (Fig. 1A). Summary clinical characteristics of this biomarker discovery cohort are provided in Table 1. All participants were of white European ancestry, and there were no significant differences in age or sex between the three groups. The T2DKD group had significantly lower mean eGFR (30.8 ml/min/1.73 m$^2$) compared to the two control groups (T2DNRF: 76.9 ml/min/1.73 m$^2$, NDNRF: 77.6 ml/min/1.73 m$^2$). Blood pressure was tightly controlled in the T2DKD group, with diastolic blood pressure of 68 mmHg being significantly lower than participants in both the T2DNRF (78 mmHg) and the NDNRF control groups (82 mmHg). There was no significant difference in glycaemic control between the two groups of participants with diabetes (mean HbA1c 62 mmol/mol T2DKD group vs 64 mmol/mol in T2DNRF group).

### Hierarchical clustering of sRNA-seq data to identify the circulating miRNome

Following alignment to miRbase, a total of 923 mature human miRs were detected. Principal component analysis revealed that T2DKD clustered away from the NDNRF and T2DNRF (Supplementary Fig. 1A). Unsupervised hierarchical clustering was performed on the most differentially expressed mature miRs across samples (Fig. 1B)(FDR < 0.1 in any comparison). The samples divided into two main groups, with one enriched for patients with T2DKD. To identify miRs that are altered in the circulation of those with CKD, we compared the miR read counts in those with T2DKD with T2DNRF. Three miRs were found to be differentially expressed between the T2DKD and the T2DNRF groups, miR-190a-5p (log FC -2.47; FDR 0.034), miR-365b-5p (log FC 4.94; FDR 0.034) and miR-7111-5p (log FC 5.21; FDR 0.048) (Supplementary Data 1). Of these miRs, miR-190a-5p was significantly lower in patients with reduced kidney function (Supplementary Fig. 1B), with the other two miRs being increased in expression (Supplementary Data 1). miR-190a-5p was detected in all samples however, the other two miRs were

detected in ~50% of samples and were not taken forward (Supplementary Data 1).

### seNSOR CKD cohort characteristics and correlation with serum miR-190a-5p

As circulating miR-190a-5p level was reduced in patients with type 2 diabetes who had impaired kidney function, we next sought to determine whether it correlated with eGFR in patients with a diverse aetiology of CKD and whether it could predict kidney outcomes. To address this, we utilised a prospective cohort of 549 patients who attended kidney outpatient clinics (seNSOR study). We excluded 135 participants who were already receiving or were close to requiring renal replacement therapy (RRT: eGFR of < 20 ml/min/1.73 m$^2$ at recruitment). In addition, 6 participants were excluded due to missing data (3 participants had no baseline eGFR measurement whilst an additional 3 participants had no follow-up data available), as were 13 participants with acute kidney injury (AKI) at baseline (Fig. 2). In 298 of the remaining 395 patients, serum miR-190a-5p was measured by RT-qPCR, with expression undetectable in 97 participants. Median eGFR was lower in those with undetectable miR-190a-5p (50 ml/min/1.73 m$^2$, IQR 31-79 vs 61 ml/min/1.73 m$^2$, IQR 37-93, $p$-value = 0.018, see Supplementary Table 2 for the detailed clinical characteristics of this group).

Clinical characteristics for participants with detectable serum miR-190a-5p are shown in Table 2, stratified according to whether their serum miR-190a-5p level was above or below the median. Participants with levels below the median tended to be older and had lower eGFR compared to those above median levels (median eGFR 55 ml/min/1.73 m$^2$ (IQR 34-89 ml/min/1.73 m$^2$) vs 67 ml/min/1.73 m$^2$ (IQR 39-99 ml/min/1.73 m$^2$), $p = 0.052$). Serum miR-190a-5p levels correlated weakly with eGFR (rho = 0.12, $p = 0.04$) and inversely with age (rho = -0.12, $p = 0.04$) but were not significantly correlated with systolic or diastolic blood pressure nor ACR (Supplementary Fig. 2).

### Low serum miR-190a-5p level predicts risk of CKD progression

During a median follow-up of 3.9 years, 70 of the 298 participants from the seNSOR cohort with detectable serum miR-190a-5p met the composite CKD progression endpoint, with 21 reaching ESKD. In those with no albuminuria (ACR < 3 mg/mmol) or moderate albuminuria (ACR < 3-300 mg/mmol), a serum miR-190a-5p level below the median was associated with increased risk of CKD progression ($p = 0.028$ in those with ACR < 3 mg/mmol and $p = 0.049$ in those with ACR 3-300 mg/mmol, Fig. 3A). However, in participants with ACR > 300 mg/mmol, serum miR-190a-5p was not predictive of CKD progression.

We proceeded to examine whether miR-190a-5p levels predicted CKD progression in univariate and multivariate Cox proportional hazards models. As shown in Table 3, in those with baseline ACR < 300 mg/mmol, miR-190a-5p was associated with CKD progression independent of age, sex, baseline systolic/diastolic blood pressure, eGFR and ACR (adjusted HR 0.80, 95% CI: 0.66-0.96, $p = 0.015$). Importantly, addition of miR-190a-5p to a model using existing clinical risk predictors resulted in an improvement in the C-statistic from 0.719 to 0.738, with the likelihood ratio test confirming a statistically significant improvement in kidney outcome prediction ($p = 0.014$). Notably, when known biomarkers urinary Epidermal Growth Factor (EGF) and Kidney Injury Molecule-1 (KIM-1) were included in this model, neither predicted CKD progression, whilst miR-190a-5p retained statistical significance (Supplementary Table 3).

When participants with an ACR > 300 mg/mmol were included in the analysis (see Supplementary Table 4), miR-190a-5p was no longer predictive of CKD progression in univariate and multivariate models (univariate HR 0.89, 95% CI: 0.77–1.02, $p = 0.087$, multivariate HR 0.86, 95% CI: 0.73–1.01, $p = 0.064$). Secondary analysis examining ESKD alone as the endpoint determined that serum miR-190a-5p level was independently associated with reaching ESKD, regardless of whether

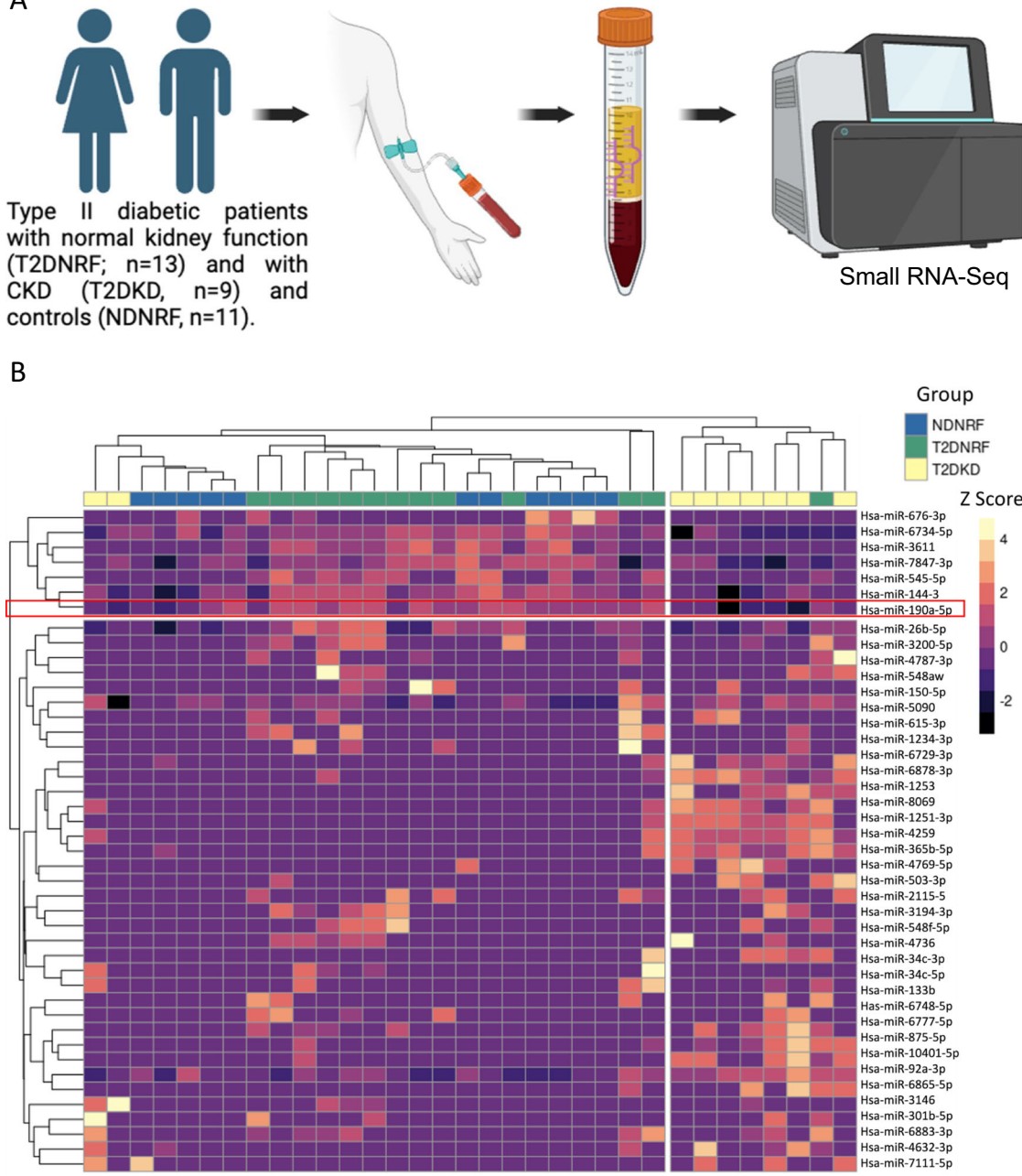

**Fig. 1 | Identification of miR-190a-5p as a differentially expressed circulating biomarker in T2D patients with normal and kidney disease. A** Schematic of the experimental approach to identify differentially expressed circulating miRNAs in patients with Type II diabetes with and without kidney disease compared to age-matched controls. BioRender. Denby, L. (2025) https://BioRender.com/z8pwjvs **B** Heatmap representation of the unsupervised clustering of miRs with an FDR < 0.1 in any comparison (n = 45) from the unbiased small RNA sequencing of plasma of patients with type 2 diabetes and kidney disease (T2DKD, n = 9), type 2 diabetes, with normal renal function (T2DNRF, n = 13) or non-diabetic controls with normal renal function (NDNRF, n = 11). Each column represents a sample, each row a miRNA, and the colour the relative expression (z score). Red box highlights miR-190a-5p.

participants with baseline ACR > 300 mg/mmol were included in the analysis (adjusted HR when including all participants 0.68, 95% CI: 0.53-0.89, $p$ = 0.015, Supplementary Table 5).

**Renal miR-190a-5p expression correlates with kidney function.** We next sought to determine the biological rationale for the correlation of serum miR-190a-5p levels with baseline eGFR and adverse kidney outcomes in patients with CKD. Firstly, we wished to examine the likely tissue source of circulating miR-190a-5p. As miR-190a-5p is found in the intronic region of the TALIN2 (*TLN2)* gene, and has been shown to use the host gene transcription start site[33], we assessed tissue

expression of *TLN2* as a surrogate of miR-190a-5p in a range of human tissues using publicly available datasets (GSE3526)[34]. *TLN2* expression was highest in the kidney (Supplementary Fig. 3A), suggesting that it may be the primary source of miR-190a-5p in the circulation. To validate this, we first assessed the expression of miR-190a-5p in frozen biopsy tissue from a subset of seNSOR patients who underwent renal biopsy at the time of recruitment, including patients with biopsy proven diabetic nephropathy (representing 15% of samples) (Fig. 3B). Expression of miR-190a-5p correlated significantly with circulating miR-190a-5p levels (Fig. 3C). The expression of miR-190a-5p in kidney biopsy tissue also correlated with eGFR (Fig. 3D) and was negatively

**Table 1 | Clinical and biochemical characteristics of the patients recruited for the sRNA-Seq miR biomarker discovery cohort for miR-190a-5p finding**

| Variable | Discovery Cohort | | | |
|---|---|---|---|---|
| | T2DKD (n = 9) | T2DNRF (n = 13) | NDNRF (n = 11) | p- value |
| Age (years) | 72.7 ± 7.0 | 69.1 ± 6.9 | 70.7 ± 7.1 | 0.503 |
| Sex (male/ female) | 7/2 | 8/5 | 7/4 | 0.705 |
| SBP (mmHg) | 123 ± 10 | 133 ± 17 | 138 ± 27 | 0.244 |
| DBP (mmHg) | 68 ± 4 | 78 ± 10 | 82 ± 17 | 0.049 |
| HbA1c (mmol/mol) | 61.7 ± 14.2 | 64.3 ± 18.0 | 39.5 ± 2.9 | 0.031 |
| Serum creatinine (µmol/l) | 181.1 ± 34.3 | 80.0 ± 16.1 | 80.2 ± 15.3 | <0.001 |
| eGFR (ml/min/ 1.73 m²) | 30.8 ± 7.3 | 76.9 ± 15.0 | 77.6 ± 10.1 | <0.001 |

Categorical values were assessed using a Chi-square test. For continuous values with a normal distribution, ANOVA was used to compare 3 groups. *T2DKD* – type 2 diabetic kidney disease, *T2DNRF* – type 2 diabetes, normal kidney function, *NDNRF* – non-diabetic, normal kidney function.

correlated with the percentage of fibrosis present in the biopsy (Fig. 3E). MiR-190a-5p expression in the biopsy also correlated with kidney tubular cell mass as assessed by immunofluorescence staining for the epithelial markers CD10 and PanCK (Fig. 3F).

We next wished to determine whether the loss of miR-190a-5p expression in the kidney was observed in pre-clinical models of kidney injury. miR-190a-5p expression in the renal cortex was significantly reduced in AKI-to-CKD transition induced by uIRI injury with nephrectomy[35] (Fig. 4A) and in progressive kidney disease induced by the subtotal nephrectomy model[36] (Fig. 4B).

Taken together, these data suggest that the level of circulating miR-190a-5p may reflect the expression in the kidney, and loss of miR-190a-5p is a feature of reduced kidney function.

**miR-190 is highly expressed in proximal tubules.** As we observed a correlation of tissue miR-190a-5p expression with tubular mass, we wished to identify the specific cellular source of miR-190a-5p within the kidney. As miR-190a-5p is conserved across species (Supplementary Fig. 3B), we employed data from a murine model of reversible unilateral ureteric obstruction (rUUO)[37,38]. Standard gene expression analysis and sRNA-seq was performed on bulk cortical tissue and on four individual cell populations isolated by fluorescence- activated cell sorting: proximal tubular cells (PT, Lotus lectin⁺), macrophages (F4/80^Hi), endothelial cells (CD31⁺) and stromal cells (PDGFRB⁺)[37]. miR-190a-5p was highest in PT cells (Fig. 4C), and was reduced following UUO before being partially restored during the repair phase (Fig. 4D), mirroring that of *Tln2* expression (Supplementary Fig. 3C). Indeed, *Tln2* expression correlated with miR-190a-5p in the rUUO model ($r = 0.9127$; $p < 0.001$) (Fig. 4E). This suggests that, in mouse and humans, *Tln2* expression could potentially act as a surrogate for miR-190a-5p expression.

**Restoring miR-190a-5p abundance in the kidney ameliorates tubular injury and renal fibrosis.** We next sought to determine whether restoring miR-190a-5p abundance in the kidney could mitigate tubular injury and maintain tubular cell health. Our previous work and that of others, has demonstrated avid uptake of miRNA-based therapies by kidney tubules[10,17,18], therefore, we administered miR-190a-5p mimic or mimic control subcutaneously in the uIRI model of AKI-to-CKD transition (Fig. 5A). We found that 14 days post-IRI there was a significant increase in miR-190a-5p expression in the kidneys of miR-190a-5p mimic treated animals compared to kidneys from mimic control-

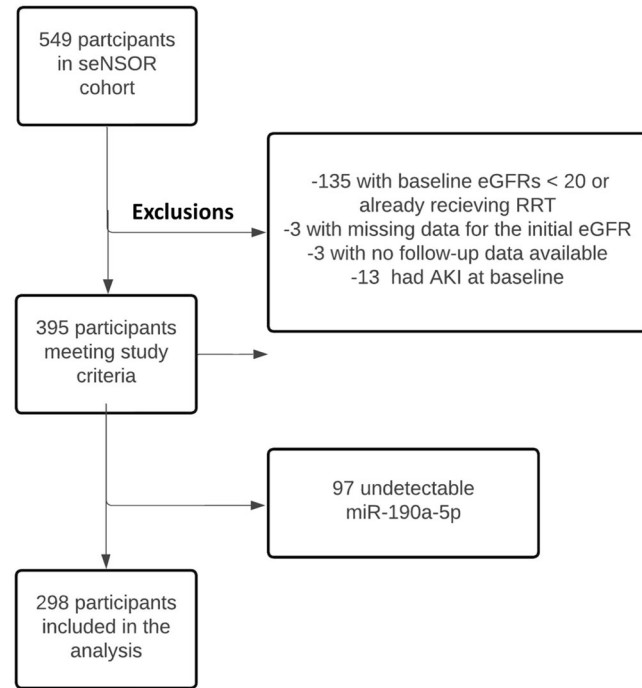

**Fig. 2 | Flowchart of patients from the seNSOR study cohort who were included and excluded from the analysis of miR-190a-5p as a potential biomarker.** From the 549 participants in the seNSOR cohort, patients were excluded who were already receiving, or close to requiring RRT (eGFR of < 20 ml/min/1.73 m² at recruitment). Patients with missing data or acute kidney injury at baseline were also excluded. From the 395 participants meeting the study criteria, miR-190a-5p was not detected in the serum from 97 patient samples. Analysis was performed in the remaining 298 participants. RRT – renal replacement therapy.

treated animals (Fig. 5B), which had miR-190a-5p expression similar to that observed in kidneys with damage induced by IRI + Nephrectomy or STNx (Fig. 4A, B). The increased renal miR-190a-5p abundance resulted in a significant decrease in *Havcr1* (encodes kidney injury molecule-1, KIM-1) expression and an increase in *Egf* and *Ass1* expression, markers of healthy Loop of Henle and proximal tubules, respectively (Fig. 5C). These findings are indicative of miR-190a-5p having a potential role in maintaining tubular health. As miR-190a-5p expression negatively correlated with fibrosis in the human biopsies, we sought to determine if preventing the loss of miR-190a-5p would have an effect on the induction of fibrosis in the uIRI murine model. We found that administration of miR-190a-5p mimic reduced fibrosis-related gene expression (Fig. 5D) and collagen deposition, as measured by PSR staining (Fig. 5E).

**Adam10 may be a miR-190a-5p target gene.** To investigate which genes are being modulated by the loss of miR-190a-5p in the PTs cells, we took advantage of the transcript sequencing data from the sorted PTs in the rUUO model and employed MultimiR to identify miR-190a-5p predicted target genes that were significantly negatively correlated with miR-190a-5p expression within proximal tubular cells (Fig. 6A and Supplementary Table 6). The 45 most significantly negatively correlated target genes were over-represented in multiple pathways, including wound healing, regulation of cytoskeleton and actin polymerisation/stabilisation (Fig. 6B). One of the genes most significantly negatively correlated with miR-190a-5p was *Adam10* (Fig. 6A; $p = 0.004$), which is a sheddase and has been shown to mediate kidney injury and fibrosis in pre-clinical models[39]. Using Targetscan, we confirmed that *ADAM10* has a 7A-mer 3′UTR binding site for miR-190a-5p and is therefore a logical candidate miR-190a-5p target gene for further study. Mediation of *Adam10* repression by miR-190a-5p is supported

**Table 2 | Baseline clinical and biochemical characteristics of the 298 patients from the seNSOR cohort who were included in the validation analyses, segregated according to whether their serum miR-190a-5p levels were above or below the median**

| | seNSOR cohort | miR-190a-5p below Median | miR-190a-5p above Median | p comparing above vs below median |
|---|---|---|---|---|
| N | 298 | 149 | 149 | – |
| Male/Female, n | 156/142 | 76/73 | 80/69 | 0.73 |
| Age (years), median(IQR) | 54 (41, 68) | 56 (43, 68) | 51 (38, 66) | 0.067 |
| eGFR (ml/min/1.73 m$^2$), median(IQR) | 61 (37, 93) | 55 (34, 89) | 67 (39, 99) | 0.052 |
| SBP (mmHg), median(IQR) | 130 (120, 145) | 130 (120, 145) | 130 (120, 144) | 0.63 |
| DBP (mmHg), median(IQR) | 78 (70, 82) | 78 (68, 84) | 79 (70, 82) | 0.81 |
| ACR (mg/mmol), median(IQR) | 19 (3, 140) | 15 (1, 141) | 21 (3, 133) | 0.36 |
| ACR <3 mg/mmol, n (%) | 67 (22) | 36 (24) | 31 (21) | – |
| ACR 3-300 mg/mmol, n (%) | 156 (52) | 78 (52) | 78 (52) | – |
| ACR >300 mg/mmol, n (%) | 29 (10) | 13 (9) | 16 (11) | – |
| ACR unavailable, n (%) | 46 (15) | 22 (15) | 24 (16) | – |
| Diabetes prevalence, n (%) | 48 (16) | 28 (18.8) | 20 (13.4) | 0.27 |
| Type 1, n (%) | 3 (1) | 2 (1.3) | 1 (0.7) | – |
| Type 2, n (%) | 45 (15) | 26 (17.4) | 19 (12.8) | – |
| Ethnicity | – | – | – | – |
| White, n (%) | 261 (88) | 129 (87) | 132 (89) | – |
| Asian, n (%) | 14 (5) | 10 (7) | 4 (3) | – |
| Black, n (%) | 6 (2) | 3 (2) | 3 (2) | – |
| Not recorded, n (%) | 17 (6) | 7 (5) | 10 (7) | – |
| Primary kidney diagnoses grouping | – | – | – | – |
| 1. Primary Glomerulonephritis, n (%) | 85 (29) | 34 (23) | 51 (34) | – |
| 2. Interstitial Nephropathies, n (%) | 77 (26) | 38 (26) | 39 (26) | – |
| 3. Multisystem diseases, n (%) | 67 (22) | 39 (26) | 28 (19) | – |
| 4. Diabetes Nephropathy, n (%) | 18 (6) | 8 (5) | 10 (7) | – |
| 5. Other, n (%) | 14 (5) | 4 (3) | 10 (7) | – |
| 6. Not known, n (%) | 37 (12) | 26 (17) | 11 (7) | – |

Categorical values were assessed using a Chi-square test. For continuous values with a normal distribution, a t test was used to compare the 2 groups. *IQR* interquartile range, *eGFR* estimated glomerular filtration rate, *SBP* systolic blood pressure, *DBP* diastolic blood pressure, *ACR* urinary albumin:creatinine ratio.

by the significant negative correlation observed between *Adam10 and* miR-190a-5p expression in bulk cortical tissue ($r = -0.77$, $p < 0.001$) in the rUUO model (Fig. 6C).

Recent studies in kidney disease have revealed the presence of a subset of injured PT cells that express inflammatory genes and are implicated in the generation of the fibrotic niche[35,40,41]. We utilised a single-nucleus RNA-Seq (snRNA-Seq) dataset generated from the non-tumorous portion of tumour nephrectomy tissue, where the kidneys were otherwise healthy or the ureter was obstructed by tumour, leading to an inflammatory and fibrotic response in the kidney[35]. We ordered cells along a trajectory from healthy to inflammatory PT cells (Fig. 6D). As miR data is not captured by snRNA-Seq, we used *TLN2* expression as a surrogate for miR-190a-5p and found it was decreased in inflammatory PT cells compared to healthy PT cells, with *ADAM10* expression being conversely increased (Fig. 6D).

Finally, we tested if *ADAM10* expression was regulated by the manipulation of miR-190a-5p expression in vitro and in vivo. We overexpressed and knocked down miR-190a-5p in the human renal proximal tubule (RPTEC) cell line (Supplementary Fig. 4A, B). We performed a dose response study (Supplementary Fig. 4C) and determined that increasing miR-190 expression reduced *ADAM10* expression in the RPTEC cells (Fig. 6E). Conversely miR-190 knock-down resulted in an increase in *ADAM10* expression (Fig. 6E). We then tested if *Adam10* was altered in our miR-190a-5p intervention study (Fig. 5) as we would hypothesise that over-expression of miR-190a-5p should repress *Adam10* expression induced by the uIRI. Indeed, we found that *Adam10* expression was significantly ($p < 0.001$) reduced in

miR-190a-5p treated kidneys when compared to mimic control treated uIRI kidneys (Fig. 6F). Hence, we confirm that miR-190a-5p may inhibit *ADAM10* expression in tubular cells, however we cannot define if this is an indirect or direct effect on *ADAM10*.

## Discussion

While miRs have been assessed previously as non-invasive diagnostic and prognostic biomarkers of CKD, most studies have employed a single miR[22,42] or a focused panel based on existing knowledge of miR biology[23,24,26,27], with only a few assessing a larger number of miRs[43–46]. Focused approaches may not identify the optimal miR biomarker, therefore, we elected to perform an unbiased screen using sRNA-seq in plasma samples, initially in patients with diabetes, with or without CKD. Plasma miR-190a-5p levels were lower in patients with diabetes who had CKD compared to those with normal kidney function. The presence of diabetes did not impact on miR-190a-5p levels, suggesting that kidney function per se was the biggest determinant of miR-190a-5p levels. Indeed, miR-190a-5p was correlated with eGFR in a second clinical cohort (seNSOR) of patients with CKD due to diverse aetiologies, suggesting that this was not a specific feature of diabetic kidney disease, but a generic phenomenon in CKD. Our study is the first to explore the association between circulating miR-190a-5p levels and current and future kidney function.

MiRs in the circulation may be derived from multiple organs, therefore, we explored the biological rationale for the reduced miR-190a-5p level observed in the circulation of patients with CKD. *TLN2*, the host gene for miR-190a-5p, was highly expressed in the human kidney,

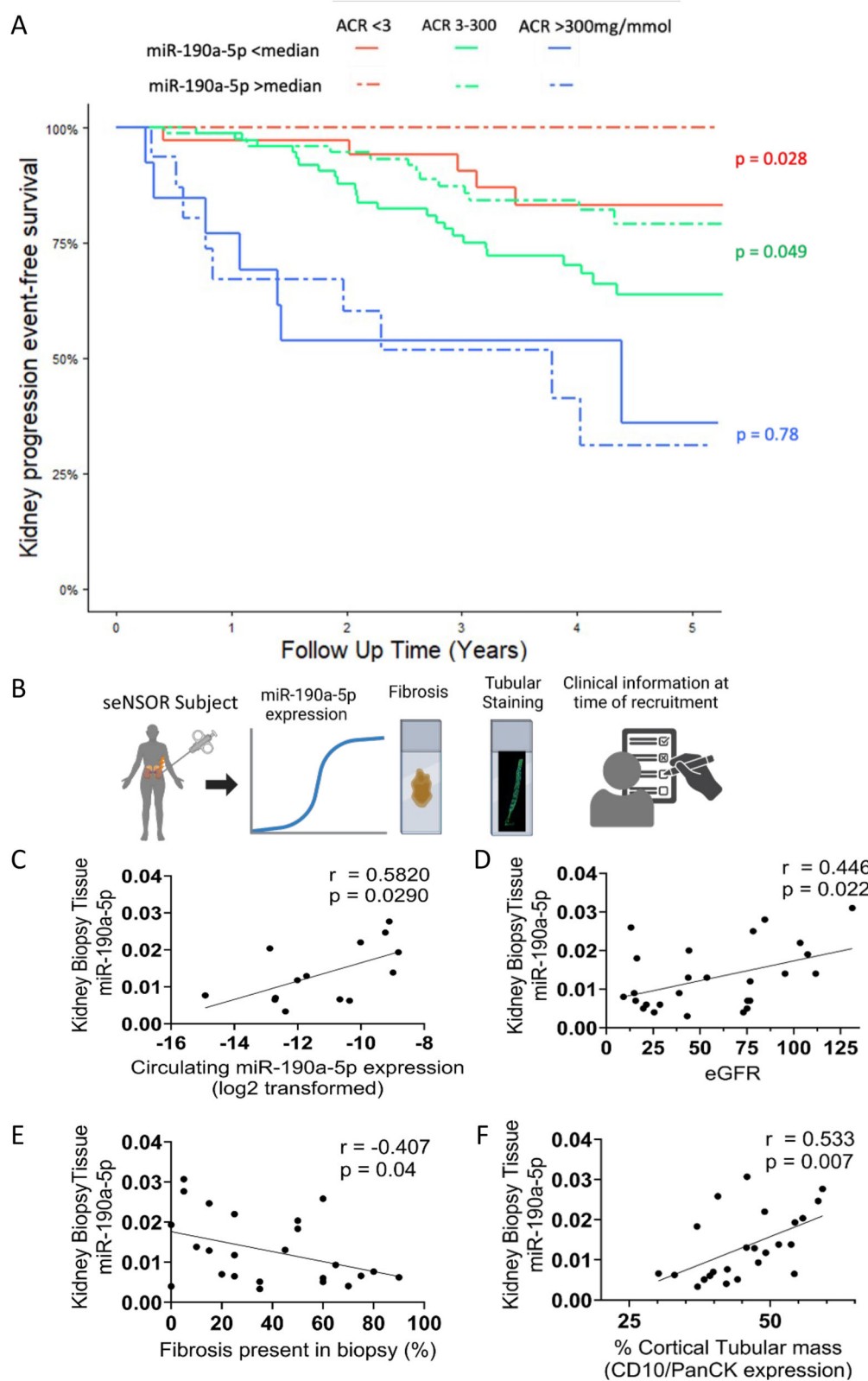

suggesting that the kidney may be the primary source of miR-190a-5p in the circulation and indeed we found that circulating levels correlated with kidney expression in a subset of our cohort. Our findings that miR-190a-5p was robustly expressed in human kidney tissue and lost as function declines, are in line with a previous report that miR-190a-5p was downregulated in kidney biopsy tissue of patients with progressive CKD compared with stable CKD[46]. Recent studies using single-cell modalities

to study the kidney in human kidney injury have identified an important inflammatory PT subcluster of the 'adaptive' tubular cells observed in the KPMP Atlas[47] that are central to the pathogenesis of kidney injury[35]. These injured PT cells are enriched for pro-inflammatory and pro-fibrotic genes and are consistent with Vcam1+ Ccl2+ PT cells which have been observed in pre-clinical models[40,41,48]. We found that *TLN2* was significantly downregulated in human inflammatory PT cells suggesting

**Fig. 3 | miR-190a-5p expression predicts kidney disease progression in patients with low to moderate proteinuria and is reduced in kidney tissue in patients with declining kidney function. A** Kaplan-Meier survival curve for progression of kidney disease. Patients within the seNSOR cohort were segregated into groups by severity of albuminuria and whether serum miR-190a-5p levels were above (dashed line) or below (solid line) the median value. ACR: urinary albumin:creatinine ratio. The Log-rank test was used to compare curves. **B** Schemata of human CKD biopsy tissue being processed for quantification of miR-190a-5p expression, fibrosis, tubular mass by CD10/PanCK staining, and linked to clinical data. Created in BioRender. Denby, L. (2025) https://BioRender.com/ee0461g (**C–F**) miR-190a-5p in renal biopsy tissue from patients within the seNSOR cohort correlates with (**C**). miR-190a-5p transcript abundance in the circulation ($n = 14$ paired samples). For the determination of linear correlation between variables, correlation coefficients (r) were generated using Pearson's test. Circulating miR-190a-5p expression was log-transformed before entering the model due to non-normal distribution (**D**). kidney function as determined by eGFR ($n = 26$ biopsy samples). Analysed by Pearson's correlation (**E**), negatively with percentage fibrosis ($n = 25$ biopsy samples). Analysed by Pearson's correlation (**F**), tubular mass as measured by CD10:PanCK staining of biopsy core ($n = 24$ biopsy samples).

**Table 3 | Univariate and multivariate Cox proportional hazards models for the risk of reaching end-stage kidney disease (ESKD) or having a sustained > 30% decline in kidney function in patients with baseline urinary ACR < 300 mg/mmol**

| | Unadjusted, univariate analysis | | | Adjusted, multivariate analysis | | |
|---|---|---|---|---|---|---|
| | HR | 95% CI | p | HR | 95% CI | p |
| log$_2$ miR-190a-5p | 0.84 | 0.72–0.98 | 0.029 | 0.8 | 0.66–0.96 | 0.015 |
| Baseline eGFR per 10 ml/min/1.73m$^2$ | 0.84 | 0.76–0.93 | <0.001 | 0.76 | 0.65–0.88 | <0.001 |
| ACR per 10 mg/mmol | 1.04 | 1.01–1.08 | 0.008 | 1.04 | 1.01–1.08 | 0.016 |
| Age per 10 years | 1.03 | 0.87–1.22 | 0.718 | 0.79 | 0.6–1.04 | 0.086 |
| SBP per 10 mmHg | 1.16 | 1.01–1.33 | 0.038 | 1.12 | 0.96–1.32 | 0.155 |
| DBP per 10 mmHg | 1.04 | 0.82–1.34 | 0.728 | 0.97 | 0.73–1.29 | 0.831 |
| Female (vs male) | 0.74 | 0.43–1.27 | 0.274 | 1.11 | 0.56–2.17 | 0.77 |

Cox univariate and multivariate proportional hazards survival models were performed to assess factors that predicted the endpoints, for multivariate analysis we excluded those patients with no ACR available.

*eGFR* estimated glomerular filtration rate, *ACR* urinary albumin:creatinine ratio, *SBP* systolic blood pressure, *DBP* diastolic blood pressure.

that miR-190a-5p may also be downregulated in this cell type. Furthermore, within the kidney, we have now demonstrated loss of miR-190a-5p in kidney biopsies in patients with CKD, enrichment of miR-190a-5p within proximal tubular cells and that kidney injury downregulates both cortical and proximal tubular miR-190a-5p expression in pre-clinical models. Furthermore, miR-190a-5p expression correlated with eGFR, tubular mass and was negatively correlated with renal fibrosis. Thus, we speculate that the reduced circulating levels of miR-190a-5p in patients with CKD may be a biomarker of reduced tubular mass and could be used to identify patients who may benefit from miR-190a-5p mimic therapy.

Importantly, low serum miR-190a-5p levels predicted adverse kidney outcomes in patients with CKD who had low-moderate proteinuria independent of traditional clinical risk factors, suggesting that the addition of miR-190a-5p may provide additional prognostic information. However, there are important caveats to its clinical use. Firstly, it is found at low levels in the circulation, particularly in patients with CKD (median 20 reads/million in those with CKD in the sRNA-seq), therefore, it might be challenging to quantify by RT-qPCR; indeed, we could not reliably detect miR-190a-5p in the serum of all the patients, typically those with lower eGFR. This may reflect the lack of tubular mass and hence more sensitive assays to detect miR-190a-5p are required to enable translation into the clinic. Secondly, miR-190a-5p levels did not provide useful prognostic information in those patients with very high levels of proteinuria. It is possible that this reflects altered binding to serum protein or increased permeability to miRs at a damaged glomerular filtration barrier; however, there was no correlation between serum miR-190a-5p levels and the severity of proteinuria. A more likely explanation is that those with heavy proteinuria have predominantly glomerular disease, and therefore a glomerular biomarker such as albuminuria will more accurately predict kidney outcome than a surrogate marker of tubular mass, such as miR-190a-5p. Importantly, serum miR-190a-5p levels will be most useful as a prognostic biomarker in those with no or low levels of proteinuria, for whom risk prediction is currently more challenging.

We provide new insight into the mechanistic role of miR-190a-5p in the kidney. It's predicted target genes are enriched in pathways including response to wounding and actin cytoskeletal organisation, and indeed its host gene, *TLN2*, is a cytoskeletal protein important for cell adhesion and motility[49]. The results from our murine studies suggest that miR-190a-5p is enriched in healthy tubular cells, where it may regulate the cytoskeleton to maintain an epithelial phenotype; indeed, miR-190a-5p has been shown to prevent epithelial-mesenchymal transition[50]. miR-190a-5p was also expressed at a much lower level in other kidney cells, including macrophages, fibroblasts and endothelial cells, therefore, we could not exclude that it may regulate gene expression in a more subtle manner in these cell types.

We demonstrate that *Adam10* expression changes reciprocally to that of miR-190a-5p in the pre-clinical model of uIRI, with supplemented miR-190a-5p expression profoundly reducing *Adam10* expression in vivo and in vitro, suggesting that *ADAM10* may be a direct or indirect target of miR-190a-5p. From our data, we postulate that the loss of miR-190a-5p in kidney disease may lead to aberrant expression of *ADAM10*, but further study would be required to provide additional evidence to support this. Further experiments would be required to demonstrate if *ADAM10* is a direct target of miR-190a-5p, such as employing a 3'UTR binding assay. Our pre-clinical data suggests that miR-190a-5p may maintain kidney cell health in part by inhibiting *ADAM10* expression, and this fits with other data which has demonstrated that *Adam10* knockout protects against kidney injury[39]. However, this requires confirmation in additional studies, such as by demonstrating that the protective effect of the miR-190a-5p mimetic is abrogated in *Adam10* knockout mice, and that conditional tubular knockout of miR-190a exacerbates kidney injury through elevated ADAM10 expression. ADAM10 promotes shedding of ectodomains of cell surface adhesion molecules such as cadherins, which are important in maintaining the integrity of the epithelial cell monolayer[51]. In addition, ADAM10 activates Notch signalling, which is important in embryological development of the kidney, however, sustained activity may promote kidney fibrosis[51]. Furthermore, *ADAM10* is up-regulated

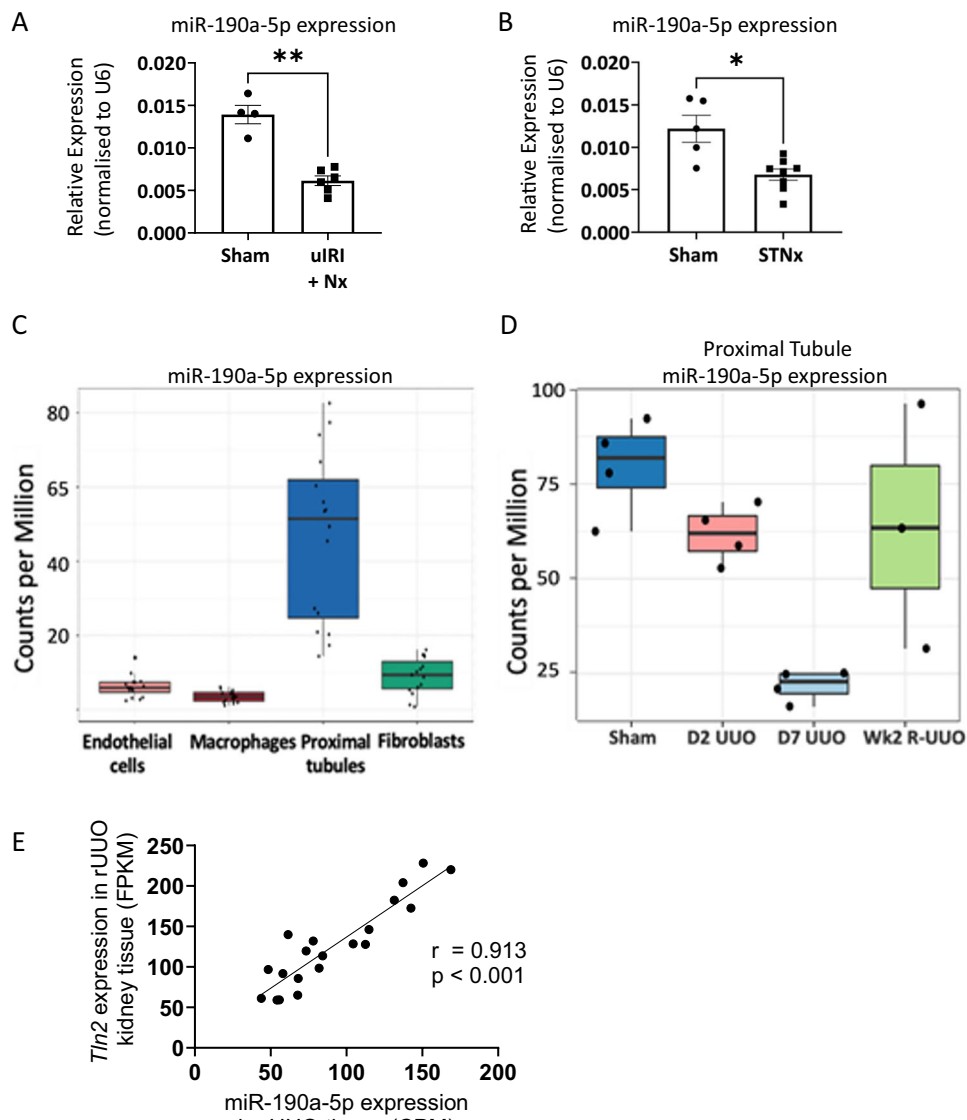

**Fig. 4 | miR-190a-5p is lost in pre-clinical models of kidney disease and is enriched in proximal tubular cells. A** miR-190a-5p expression was measured by RT-qPCR, normalised to U6 in RNA extracted from kidneys from mice culled 2 weeks after sham ($n = 4$, male mice) or uIRI surgery with contralateral nephrectomy (Nx, $n = 6$, male mice). Data are presented as mean normalised expression +/− SEM. Data analysed by Mann-Whitney test. **$p = 0.0095$ vs sham animals (**B**) miR-190a-5p expression was measured by RT-qPCR, normalised to U6 in RNA extracted from kidneys from mice culled 12 weeks post sham ($n = 5$, male) or subtotal nephrectomy (STNx) surgery to induce progressive kidney disease ($n = 8$, male). Data are presented as mean normalised expression +/− SEM. Data analysed by Mann Whitney test. *$p = 0.0186$ vs Sham animals. **C** Expression of miR-190a-5p determined by

small RNA-seq performed on cell subsets isolated by FACS from the renal cortex of mice undergoing a reversible ureteric obstruction model. $n = 16$/cell type. Box (IQR) is the median, Q1 (25th percentile), Q3 (75th percentile). The whiskers represent: Minimum Q1-1.5*IQR and Maximum Q3-1.5*IQR. IQR = Interquartile range. **D** Small RNA-Seq of sorted cortical proximal tubular cells during the rUUO model. $n = 3$-4/gp. Box (IQR) is the median, Q1 (25th percentile), Q3 (75th percentile). The whiskers represent: Minimum Q1-1.5*IQR and Maximum Q3-1.5*IQR. IQR = Interquartile range (**E**). Correlation of miR-190a-5p expression and *Tln2* in bulk cortical rUUO tissue. $n = 20$. Analysed by Pearson's correlation. FPKM = Fragments Per Kilobase of transcript per Million mapped reads. CPM = Counts per Million.

by TGF-β to act as the principal ephrin-B2 sheddase in fibroblasts, and pharmacological inhibition of ADAM10 inhibits the development of pulmonary fibrosis[52].

In addition to its role as a biomarker, miR-190a-5p offers promise as a therapy. Over-expression of miR-190a-5p in the uIRI model had a pronounced effect, reducing tubular injury and fibrosis, and preserving markers of differentiated tubular cells. Others have reported that miR-190a-5p inhibits epithelial-mesenchymal differentiation in cancer cells, reducing tumour growth and metastases in in vivo models[50,53]. In addition, miR-190a-5p promotes tissue adaptation to hypoxia[54], protects against oxidative stress[55], and miR-190 mimics ameliorated neurological injury in pre-clinical models of ischaemic stroke[56]. Our approach to overexpress miR-190a-5p was

untargeted, with quite wide variation in the level of miR-190a-5p overexpression achieved in the kidney using this approach. There is likely scope to improve the efficacy using a more refined kidney targeting methods, such as kidney-targeted lipid nanoparticles[57], with improved payload delivery[58].

In summary, we have used sRNA-seq to identify that miR-190a-5p levels are reduced in patients with lower kidney function and that low circulating miR-190a-5p can predict adverse kidney outcomes in patients with low-grade proteinuria independently of standard risk factors, though these findings require replication in additional cohorts. miR-190a-5p is predominantly expressed in healthy kidney tubules and it may regulate *ADAM10* expression and miR-190a-5p mimics should be explored as a potential therapy for CKD.

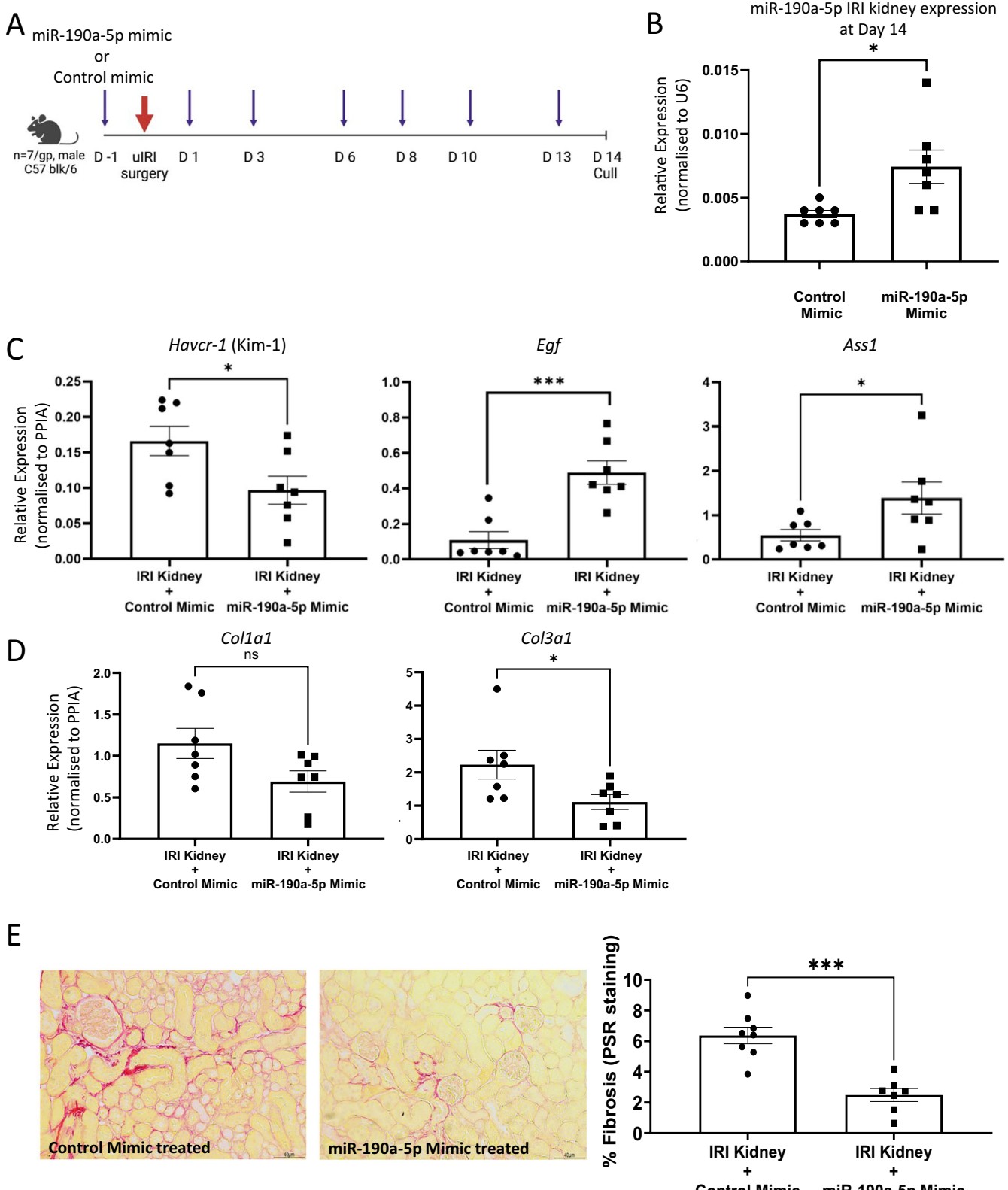

**Methods**

**Study populations**

For biomarker discovery using small RNA-sequencing, participants were recruited and gave written consent from kidney and cardiovascular clinics at the Belfast City and Royal Victoria Hospitals, Belfast. At the time of the study visit, the presence of CKD was determined on the basis of ≥ 2 eGFR measurements < 60 ml/min/1.73 m² > 3 months apart or persistent albuminuria expressed as albumin-to-creatinine ratio (ACR) ≥ 3 mg/mmol in line with the clinical definition of CKD[6]. T2DM was diagnosed according to the American Diabetes Association 2010 criteria[59]. Patients with systemic metabolic disease other than T2DM, or those who had experienced infection within the previous month, were excluded from the study. In the discovery cohort (n = 33) patients were categorised into 3 groups: type 2 diabetes with kidney disease

**Fig. 5 | Overexpression of miR-190a-5p in the kidney reduces injury and fibrosis induced by uIRI. A** Schema of the intervention study in which mice were randomly assigned to either receive subcutaneous treatment with miR-190a-5p mimic or control mimic ($n$ = 7/gp) and uIRI surgery. Animals were dosed on 7 occasions (purple arrows) and culled at Day 14 post-surgery. Created in BioRender. Denby, L. (2025) https://BioRender.com/s67k0ep (**B**) miR-190a-5p expression in the kidney following intervention with miR-190a-5p or control mimic was determined by RT-qPCR from RNA extracted from the IRI kidneys and normalised to U6 ($n$ = 7/group). Data analysed by an unpaired Student's $t$ test. *$p$ = 0.0167 vs control mimic treated. **C** RNA extracted from the IRI kidneys at Day 14 was profiled for the expression of genes associated with tubular health and normalised to *Ppia* ($n$ = 7/group). Data analysed by an unpaired Student's $t$ test. For *Havcr-1* *$p$ = 0.0322; *Egf* ***$p$ = 0.0002; *Ass1* *$p$ = 0.0489 vs control mimic treated. **D** Gene expression of fibrosis-related transcripts were assessed in RNA extracted from the IRI kidneys at Day 14 and normalised to *Ppia* ($n$ = 7/group). Data analysed by unpaired Student's $t$ test. *$p$ = 0.0388 vs control mimic treated. **E** Fibrosis was semi-quantified by picrosirus red staining of non-overlapping FFPE sections prepared from IRI and sham kidneys at Day 14 ($n$ = 7/group). Staining was quantified using ImageJ. Data analysed by an unpaired Student's $t$ test. *** $p$ = 0.0003 vs control treated kidney.

($n$ = 9) (T2DKD - T2DM and eGFR < 60 ml/min/1.73 m², ACR in 3–30 mg/mmol range), type 2 diabetes and normal kidney function ($n$ = 13) (T2DNRF - T2DM and eGFR > 60 ml/min/1.73 m², ACR ≤ 3 mg/mmol) and no diabetes and normal kidney function ($n$ = 11)(NDNRF).

To assess the utility of miR-190a-5p as a prognostic biomarker in patients with unselected CKD, we employed 549 patients from a prospective cohort study – 'Non-invasive biomarkers of kidney disease' (seNSOR), which recruited patients with unselected aetiology of CKD from outpatient clinics at the Royal Infirmary, Edinburgh, Scotland between March 2017 to March 2019. Participants with acute kidney injury (AKI) determined using the KDIGO creatinine-based criteria[6] were excluded.

In both centres, research nurses obtained written consent and participant information was collected upon enrolment, including age, self-reported sex, ethnicity, diabetes status, blood pressure, aetiology of CKD and current prescribed medications. Baseline laboratory data obtained included: serum creatinine, albuminuria (as urinary albumin to creatinine ratio, uACR) and glycated haemoglobin A1c (HbA1c). Glomerular filtration rate was estimated from serum creatinine using the 2009 Chronic Kidney Disease Epidemiology Collaboration (CKD-EPI) equation, excluding race[60]. For the seNSOR cohort, kidney biopsy data and kidney outcome data were captured using the NHS Lothian patient record systems. The primary outcome of CKD progression was defined as reaching ESKD (starting renal replacement therapy (RRT) or maintaining an eGFR < 15 ml/min/1.73 m² for > 90 days, or > 30% reduction in kidney function from eGFR at baseline maintained for > 90 days). Reaching ESKD alone was also used as a secondary outcome. Ethical approval was obtained from the respective Offices for Research Ethics Committees (Northern Ireland REC/14/NI/1132; Scotland REC/15/ES/0094). For the biopsy samples, tissue use was approved by the steering committee of the National Research Scotland Lothian Bioresource (REC 20/ES/0061, study SR2175).

### Small RNA-sequencing (sRNA-Seq)

Whole blood was collected in an ethylenediaminetetraacetic acid (EDTA) collection tube, centrifuged at 3000 × $g$ for 10 minutes at 4 °C, and the supernatant aliquoted in RNase/DNase-free tubes and frozen at − 80 °C. RNA was extracted from 600 μl of plasma using the Next-Prep™ MagnaZol™ cfRNA Isolation Kit (Bio Scientific Corp, Texas, USA) and quantified using the Qubit miRNA Assay kit (Thermo Fisher Scientific, San Jose, USA).

microRNA libraries were prepared using QIAseq miRNA Library Kit (Qiagen, Hilden, Germany). Library concentrations were quantified using the Qubit dsDNA HS Assay Kit (Thermo Fisher Scientific), and quality control was performed using a High Sensitivity NGS Fragment Analysis Kit (Agilent Technologies, Santa Clara, CA, USA). Libraries were pooled at 4 nM molarity and sequenced using the NextSeq 500 High-Output Kit (Illumina, San Diego, CA, USA). Raw read quality control was performed with FastQC, adaptors trimmed with Trim Galore, reads collapsed with seqcluster and aligned with Bowtie1 against miRBase mature miRNA and miRBase hairpin (miRbase Ver 22, available at http://mirbase.org). Post-alignment processing of miRBase hairpin was performed using SAMtools and EdgeR (3.36.0) for miRBase

analysis. The FASTQ files and expression matrix are available in Gene Expression Omnibus (GEO) (GSE262414). This provided a count table of miRNA reads ready for downstream analysis in RStudio (Ver 4.1.1). As previously performed[37], exploratory principal component analysis was first performed using DESeq2 (Version 1.34.0). After count normalisation and removal of lowly expressed reads, differential expression analysis was performed using EdgeR (Version 3.36.0). A fold change > 1.5 and false discovery rate (FDR, Benjamini-Hochberg method) < 0.05 was considered significant.

### RNA-Sequencing of bulk cortical tissue and fluorescence-activated cell sorting of specific kidney cell types.

As previously described[37,38], male 8–12-week-old C57BL/6 mice (Envigo) ($n$ = 5/group) underwent laparotomy and unilateral ureteric obstruction (UUO) followed by ureteric reimplantation to reverse the obstruction for the rUUO group. Mice were culled at day 2 after UUO (UUO-2), at day 7 after UUO (UUO-7), and in the rUUO group at 14 days after reversal by exsanguination under terminal isoflurane anaesthesia. Immediately after culling, mice were perfused with 5 mL PBS. The kidneys were harvested, with the kidney capsule removed and placed in ice-cold PBS and processed as previously described[37,38]. RNA integrity was checked using Agilent Technologies picochips. All samples utilised for sRNA-Seq had a minimum RNA integrity score of 7. For the bulk and single-population sRNA-Seq, RNA underwent sRNA-Seq by Genewiz, using the Illumina HiSeq platform, generating paired-end reads of 50 bp ($n$ = 4 per group)(sRNA-Seq dataset - GSE150035). For gene expression analysis of sorted cell populations, RNA integrity was ≥ 9 for all samples, amplified cDNA was prepared using Ovation® RNA-Seq System V2 (NuGEN) and sequenced by Genewiz on the Illumina HiSeq platform with 2x150bp configuration ($n$ = 4 per group). Data from the gene expression from sorted cell populations is available at GEO (GSE262799).

Gene targets of miR-190a-5p were identified using the R package multiMiR (1.6.0). multiMiR allows the user to identify validated miRNAs from multiple databases. For predicted miR-gene interactions, we selected only interactions which were conserved and present in more than one database. We performed pairwise correlation of proximal tubular cell expression changes (GSE262799) of predicted miR-190a-5p gene targets. Genes negatively correlated to miR-190a-5p (correlation coefficient of < − 0.6) were considered potential gene targets. Finally, the pathways enriched for miR-190a-5p target genes were explored using the package ClusterProfiler (3.12.0).

### Analysis of miR-190a-5p in seNSOR samples.

Whole blood was collected into a BD Vacutainer® SST™ Tubes, centrifuged at 3000 × $g$ for 10 min at 4 °C, and the supernatant was aliquoted in RNase/DNase-free tubes and frozen at -80 °C. Synthetic *C.elegans* miR-39 (Qiagen) was added as an exogenous reference miR prior to RNA extraction from 200 μl of serum using the miRNeasy serum/plasma kit (Qiagen). Reverse transcription was performed using the micro-RNA reverse transcription kit (Thermo Fisher Scientific) with specific primers for hsa-miR-190a-5p and cel-miR-39. Real-time quantitative

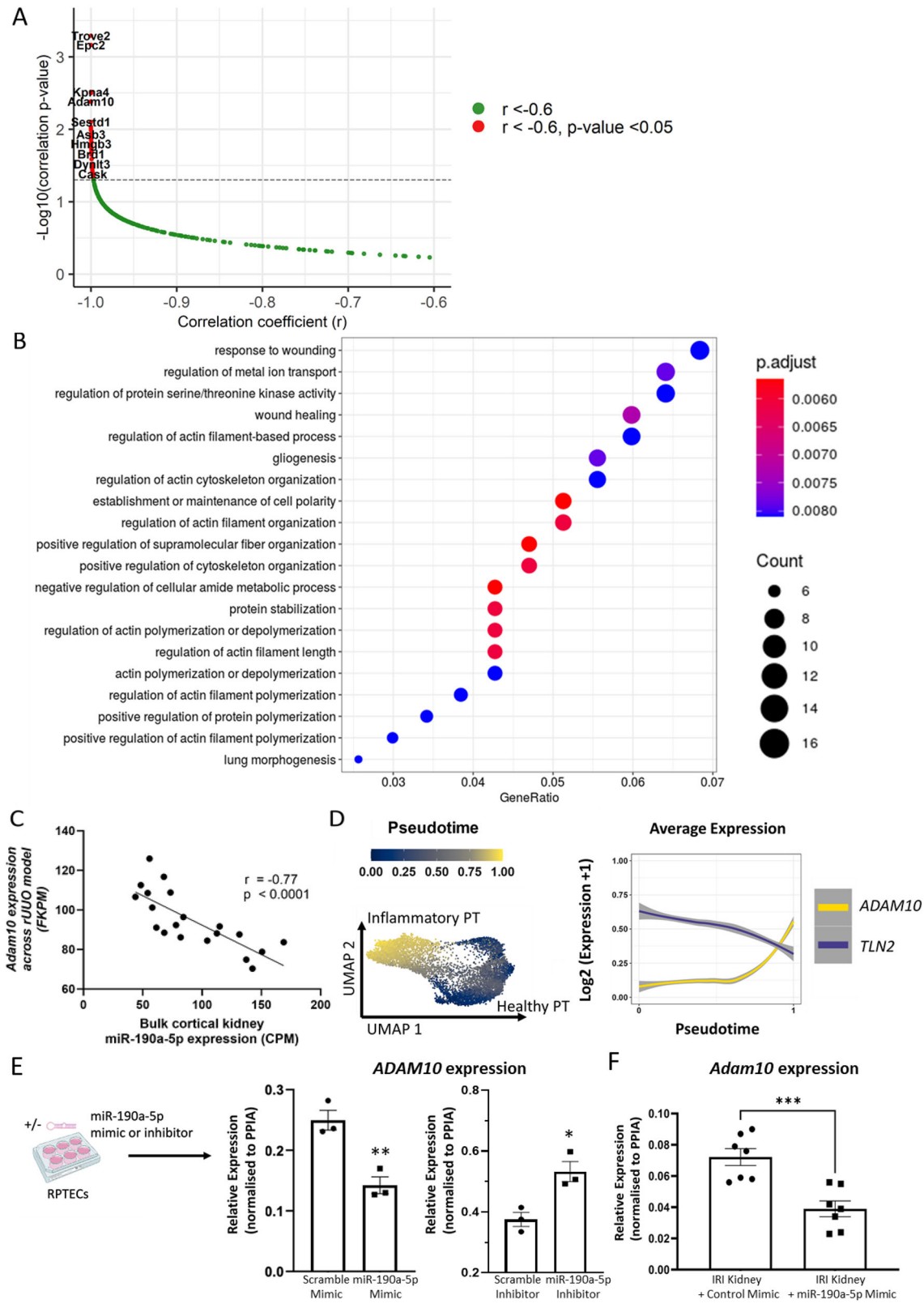

PCR (RT-qPCR) was performed using specific primer:probes (Thermo Fisher Scientific) (Supplementary Table 1) with Universal Master Mix II (Thermo Fisher Scientific) on the QuantStudio 5 (Thermo Fisher Scientific). Cycle threshold was obtained for miR-190a-5p and normalised to cel-miR-39, and relative miRNA expression calculated using the $2^{-\Delta CT}$ method and relative standard error of the mean plotted.

For biopsy samples, a subset of seNSOR patient biopsies samples, frozen in optimal cutting temperature (OCT) compound, were obtained from the NHS Lothian Biobank (REC 20/ES/0061, studies SR2175) ($n = 26$). RNA was extracted and reverse transcription performed using the microRNA reverse transcription kit (Thermo Fisher Scientific) with specific primers for hsa-miR-190a-5p and control snoRNA RNU48 (Supplementary Table 1). RT-qPCR was performed

**Fig. 6 | Identification and validation of *ADAM10* as a miR190a-5p target in proximal tubular cells. A** Analysis of potential negatively correlated gene targets of miR-190a-5p ordered by correlation coefficient (r) and −log10(*p*-value) in PT cells from the rUUO model. r < -0.6, *p*-value < 0.05. Analysed by Pearson's correlation. PT = Proximal Tubule (**B**). Pathway enrichment analysis for negatively correlated miR-190a-5p target genes. **C** Correlation of *Adam10* with miR-190a-5p expression in the renal cortex of the reversible ureteric obstruction (rUUO) model. For the determination of linear correlation between variables, correlation coefficients (r) were generated using Pearson's test. FPKM = fragments per kilobase mapped. CPM = Counts per million. **D** PT cells ordered in pseudotime trajectory from healthy to inflammatory cell state in the non-tumorous portion of nephrectomy tissue from patients with and without ureteric obstruction. Right graph. Mean *ADAM10* or *TLN2* expression in each cell along the pseudotime trajectory. **E** *ADAM10* expression in human proximal tubular cell line (RPTECs) with over-expression (50 nM mimic) or knockdown (100 nM inhibitor) of miR-190a-5p. Data analysed by unpaired Student's *t* test. **\*\****p* = 0.0073 (mimic), *\*p* = 0.0173 (inhibitor) vs scrambled control (*n* = 3 biological replicates). PT – proximal tubule Created in BioRender. Denby, L. (2025) https://BioRender.com/0ljj331 (**F**). *Adam10* expression in the renal cortex of mice that underwent ischaemia-reperfusion injury and were administered miR-190a-5p mimic or scrambled control (*n* = 7/group). Data analysed by an unpaired Student's *t* test. **\*\*\****p* = 0.0007 vs Control Mimic.

using specific primer:probes (Thermo Fisher Scientific)(Supplementary Table 1) with Universal Master Mix II (Thermo Fisher Scientific) on the QuantStudio 5 (Thermo Fisher Scientific). Cycle threshold was obtained for miR-190a-5p and normalised to RNU48, and relative miRNA expression calculated using the $2^{-\Delta CT}$ method and relative standard error of the mean plotted.

**Histological analysis of seNSOR biopsies.** Immunofluorescence staining on 24 human kidney biopsy samples was performed using a BOND III automated immunostainer (Leica Biosystems). Pancytokeratin (PanCK, Merck, Cat C2562) at 1:6000 dilution, and CD10 (Leica, Cat: NCL-L-CD10-270) at 1:600 were added together, and the same Opal 520 fluorophore used for both, so together they represent pan-tubular epithelial markers. Images of whole slides were acquired using the Axio Scan Z1 whole slide scanner (Zeiss, Jena, Germany). Images were then analysed using QUPath (version 0.5.1) and areas of cortex manually annotated. Areas of positive staining for PanCK / CD10 were expressed as a percentage of the cortical tissue to calculate tubular mass percentage.

Cortical slides stained for Masson's Trichrome were available for 25 biopsy samples from which miR-190a-5p expression had been determined. A single blinded operator performed semi-quantitative cortical interstitial fibrosis assessment on these slides.

**Cell culture of kidney proximal tubular cells (RPTECs)**
Human kidney proximal tubular epithelial cells (RPTEC/TERT1) (ATCC® CRL-4031™ LGC Limited, UK) were maintained in complete growth media consisting of DMEM/F-12 (2.5 mM L-glutamine, 15 mM HEPES, 0.5 mM sodium pyruvate, and 1200 mg L$^{-1}$ sodium bicarbonate) (Gibco, New York, USA), geneticin 2 μL mL$^{-1}$ (Gibco, New York, USA), hydrocortisone 25 ng mL$^{-1}$ (Sigma Aldrich, Gillingham, UK), ascorbic acid 3 μg mL$^{-1}$ (Sigma-Aldrich, Gillingham, UK), sodium selenite 6.7 ng mL$^{-1}$ transferrin 5.5 μg mL$^{-1}$, insulin 10 μg mL$^{-1}$ (Gibco, New York, USA), triiodo-L-thyronine 6 pM (Sigma-Aldrich, UK), prostaglandin E1 25 ng mL$^{-1}$ (Sigma-Aldrich, Gillingham, UK), rhEGF 10 ng mL$^{-1}$ (Promega, Southampton, UK). Cells were grown at 37 °C, 5% $CO_2$, in T75 cm$^2$ flasks to 100% confluence. For passaging, cells were detached using trypsin, spun and resuspended in fresh media.

**Manipulation of miR-190a-5p levels in RPTECs**
Dharmacon miR-190a-5p miRNA mimic (C-300639-03-0002) and inhibitor (IH-300639-05-0002), along with negative control mimic (CN-001000-01-05) and miRNA hairpin inhibitor negative control (IN-001005-01-05) were purchased from Horizon Discovery. RPTECs were transfected with 100 nM of miR-190a-5p inhibitor or 5–50 nM miR-190a-5p mimic or controls using siPORT™ NeoFX™ Transfection agent in Opti-MEM for 6 h before adding complete media. Media was replaced after 24 h and cells left for a further 48 h prior to being collected in Qiazol.

RNA was extracted from cells using the miRNeasy mini kit (Qiagen) following the manufacturer's instructions. cDNA was prepared using the reverse transcription kit with random hexamers (Thermo Fisher Scientific). RT-qPCR was performed using specific primer:probes for a disintegrin and metalloproteinase domain-containing protein 10 (*ADAM10*) and peptidylprolyl isomerase A (*PPIA*) was used as the reference gene (Supplementary Table 1). Reverse transcription was performed using the microRNA reverse transcription kit (Thermo Fisher Scientific) using specific primers for miR-190a-5p and snoRNA RNU48 (Supplementary Table 1) as an endogenous reference small RNA, and RT-qPCR performed using specific primer:probes (Thermo Fisher Scientific). All RT-qPCR was performed with Universal Master Mix II (Thermo Fisher Scientific) on the QuantStudio 5 (Thermo Fisher Scientific). Relative expression was calculated as a change in cycle threshold (ΔCT) for each sample to the housekeeper and plotted as $2^{-\Delta CT}$, and the relative standard error of the mean plotted.

**miR-190a-5p expression and manipulation in vivo.** All procedures were approved by the Animal Welfare and Ethical Review Body (AWERB) at the University of Edinburgh and were conducted in accordance with the United Kingdom Animals Scientific Procedures Act 1986 and the ARRIVE guidelines under Project Licences approved by the UK Home Office. Our human data contained male and female biological sex, and the effects observed were independent of biological sex. Therefore, for our pre-clinical models, we used only male animals as these have been well characterised in-house and previous publications with robust data were available to perform power calculations to determine group size.

C57BL/6 J mice and SV129 mice were purchased from Enviago and housed in a pathogen-free environment at the University of Edinburgh. Mice were housed at 50% humidity and 22–26 °C with 12: 12 light: dark cycles starting at 07:00 local time. Mice were group-housed and given ad libitum access to water and standard chow (rat and mouse maintenance diet 1 (Special Diet Services)).

Subtotal nephrectomy was performed as previously described on SV129 mice[36]. Briefly, mice were randomly allocated to either receiving a sham (*n* = 5) or subtotal nephrectomy surgery (*n* = 8). Animals were recovered and left for 12 weeks to allow progressive kidney injury to occur. At 12 weeks animals were culled and tissue taken.

Unilateral IRI (uIRI) surgery was performed as previously described[35,40]. Briefly, a flank incision was made, and for sham surgery (*n* = 4), the left renal pedicle identified and touched with a moist cotton bud, for uIRI animals (*n* = 6), it was clipped using an atraumatic clamp for 18 min. During the ischemic period, body temperature was maintained at 37 °C using a heating blanket with homoeostatic control (Harvard Apparatus), measured by a rectal temperature probe. The clamp was then removed, the peritoneum closed with 5/0 suture, and the skin incision closed with clips. At 14 days post-surgery, a flank incision was made on the right side and a contralateral nephrectomy performed to produce a function model of IRI injury, and animals culled 7 days post-nephrectomy.

For the intervention study, mice (*n* = 7/gp) were assigned to either an experimental group receiving subcutaneous miR-190a-5p mimic (purchased from Horizon Discovery) injected at 0.625 μg per kg in PBS or scrambled control mimic (Horizon Discovery) using randomiser software and end analyses were conducted blind. Animals were dosed

Day -1, 1, 3, 6, 8, 10, 13 and culled on day 14, without a contralateral nephrectomy.

For all in vivo studies, kidneys were divided and one portion snap frozen for RNA extraction, and the other portion stored in 10% formalin for histological analysis.

**Histological analysis of mouse kidney tissue.** Formalin fixed and paraffin embedded (FFPE) mouse kidney tissue was sectioned at 3 µM. Sections were deparaffinized twice for 5 min in xylene and rehydrated in sequential 100%, 90% and 70% ethanol for 2 min each and placed in distilled water. Picrosirius red (PSR) staining was performed using the Picrosirius Red Stain Kit (Abcam, ab150681) following manufacturer guidelines. After rehydration, tissue sections were incubated in PSR for 60 min. Slides were rinsed in acetic acid solution and dehydrated in absolute ethanol before mounting. Non-overlapping images of sections were obtained for each animal and analysed blinded by a single operator with Fiji ImageJ to determine the percentage of fibrosis.

**Gene Expression analysis in mouse tissue.** Total RNA was extracted from snap-frozen kidney tissue using the miRNeasy Mini kit (Qiagen), following the manufacturer's instructions. cDNA for quantitative PCR was synthesised from the extracted RNA using the reverse transcription kit with random hexamers (Thermo Fisher Scientific). RT-qPCR was performed using TaqMan Universal Master Mix II (Applied Biosystems) and TaqMan Gene Expression Assay-specific primers (Supplementary Table 1) and normalised to *Ppia* expression.

### Single-cell analyses

For correlation analysis of *ADAM10* and *TLN2* single nucleus multiome (paired gene expression and chromatin accessibility) data was downloaded from GEO in Seurat (v4.4.0) format (GSE254185)[35]. Due to sparsity and technical dropouts in single-cell transcriptomics data, we opted to assess the average expression levels in partitions of the nearest neighbour graph instead of individual cells for gene-to-gene correlations.

For this purpose, using the original cell annotations, the data was subset to proximal tubule (PT) cells and subclusters of the weighted shared nearest neighbour (wsnn) graph were constructed using the FindClusters function with resolution $n = 8$ (median: 131 cells/cluster). Clusters were subsequently pruned by discarding clusters with less than 20 cells. The average log-scale expression level of *ADAM10* and *TLN2* was derived using the AverageExpression function, and cluster-level average expression was used for visualisation as a scatterplot.

### Statistical analyses

The normality of distribution for all key variables was assessed by the Shapiro-Wilk test. Clinical characteristics for continuous data are expressed as mean ± standard deviation for normally distributed data and median (interquartile range) when not normally distributed. Categorical variables were expressed as counts (percentages). For continuous values with normal distribution, t-tests or ANOVA were used to compare 2 groups or > 2 groups, respectively. For continuous values without normal distribution, the Mann-Whitney test was used to compare 2 groups and a Kruskal-Wallis test was employed if > 2 groups. Categorical values were assessed using a Chi-square test. For the determination of linear correlation between variables, correlation coefficients (r) were estimated using Spearman's rank tests (if the data was not normally distributed) or Pearson's (if the data was normally distributed). Kaplan-Meier survival curves were constructed for the primary kidney endpoints comparing above and below median miR-190a-5p separately for each uACR stage (< 3 mg/mmol, 3–300 mg/mmol, > 300 mg/mmol), with the log-rank test used to compare curves. Cox univariate and multivariate proportional hazards survival models were performed to assess factors that predicted the endpoints, for multivariate

analysis we excluded those patients with no ACR available. Circulating miR-190a-5p expression was log-transformed before entering the model due to non-normal distribution. For in vitro miR-190 mimic and inhibitor experiments, biological repeats were used ($n = 3$), with 3 technical replicates per sample and results analysed by Student's unpaired $t$ test. All tests were performed using SPSS version 28.0 (IBM, Armonk, NY), R version 4.1.2 (R Foundation for Statistical Computing, Vienna, Austria) or Graphpad Prism version 10.1.2 (Graphpad software, Boston, Massachusetts). A $p$-value < 0.05 was considered significant.

## Data availability

The raw and processed data used in this manuscript are available in Gene Expression Omnibus with the following accession numbers: GSE262414 https://www.ncbi.nlm.nih.gov/geo/query/acc.cgi?acc=GSE262414. GSE262799 https://www.ncbi.nlm.nih.gov/geo/query/acc.cgi?acc=GSE262799. GSE254185 https://www.ncbi.nlm.nih.gov/geo/query/acc.cgi?acc=GSE254185. GSE150035 https://www.ncbi.nlm.nih.gov/geo/query/acc.cgi?acc=GSE150035. Complete seNSOR clinical dataset – controlled access due to patient privacy, the de-identified data and accompanying R code used to generate figures is available by contacting the corresponding author (Laura.Denby@ed.ac.uk) and all requests will be actioned within 4 weeks of email receipt. Source data are provided in this paper.

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

## Acknowledgements

This project was funded by Kidney Research UK grants RP 042_20160304 (awarded to LD) and RP_013_20221129 (awarded to BRC), Northern Ireland Kidney Research Fund, the Northern Ireland Health and Social Care Research and Development Office (STL/4936/14) and the Chinese Scholarship Council. DB and KC were supported by MRC Clinical Training Fellowships (MR/W00089X/1 and MR/S001743/1). M.R. is supported by a PhD studentship from the MRC Precision Medicine Programme (MR/N013166/1). LD is supported by a Kidney Research UK Senior Fellowship (SF_001_20181122). RMB is supported by an MRC award MR/Y014103/1 awarded to LD. The authors would like to acknowledge the patients who agreed to take part in the research, and the research nurse teams of the Clinical Research Facilities at the Queen Elizabeth University Hospital in Glasgow and The Royal Infirmary of Edinburgh for the recruitment and collection of samples for seNSOR. In addition, we thank the staff of the Lothian and GGC biorepositories. The authors would like to acknowledge technical assistance from Sharon Alexander, Angeline McBriar and the Queen's University Belfast Genomics Core Technology Unit (available at https://www.qub.ac.uk/sites/core-technology-units/Genomics/). The authors would like to acknowledge the assistance of the Queen's Medical Research Institute Flow Cytometry and Cell Sorting Facility, University of Edinburgh. The authors would like to acknowledge Prof Chris Cardwell, Queen's University of Belfast, for statistical advice. The authors acknowledge the use of BioRender in generating figures for this paper. The authors would also like to acknowledge Richard H Croasdale for help with the setting up and maintenance of the servers of our shiny apps.

## Author contributions

Conceptualisation: J.H., P.B.M., A.P.M., G.J.M., D.A.S., B.R.C. and L.D. Methodology: D.P.B., J.Z., K.L.C., O.T., M.R., C.C., J.P.T., R.K.Y.W, P.B.M., A.P.M., G.J.M., D.A.S. and L.D. Investigation: D.P.B., J.Z., K.L.C., O.T., R.K.Y.W., M.R., C.C., C.S., R.M.B.B., J.P.T., D.A.F. and L.D. Visualisation: D.P.B., M.R., K.C., B.R.C. and L.D. Supervision: A.P.M., G.J.M., D.A.S., B.R.C. and L.D. Writing-original draft: D.P.B., J.Z., K.L.C., A.P.M., G.J.M., D.A.S., B.R.C. and L.D. Writing – review and editing: D.P.B., J.Z., K.L.C., O.T., R.K.Y.W., M.R., C.C., J.P.T., J.H., P.B.M., D.A.F., A.P.M., G.J.M., D.A.S., B.R.C. and L.D.

## Competing interests

These authors declare no competing interests: J.Z., K.L.C., O.T., R.K.Y.W., M.R., C.C., J.P.T., J.H., A.P.M., G.J.M., D.A.S. and L.D. These authors declare the following competing interests: P.B.M. reports lecture or advisory board honoraria from AstraZeneca, Bayer, Boehringer Ingelheim, Pharmacosmos, Astellas, Vifor, GSK and research funding from AstraZeneca and Boehringer Ingelheim outside the submitted work. B.R.C. reports consultancy from Argenx and research funding from AstraZeneca outside the submitted work. D.P.B. is listed as a co-inventor on a patent application covering urinary protein biomarkers in chronic kidney disease outside the submitted work.
