## [Transparent Peer Review file · Nature Communications]

Low circulating miR-190a-5p predicts progression of chronic kidney disease.

Corresponding Author: Dr Laura Denby

Version 0:

Reviewer comments:

Reviewer #1

(Remarks to the Author)

The present manuscript by Baird et al. highlights the role of miR-190a-5p in progression to chronic kidney disease. First the authors compare the expression profile of small RNAs in the circulation of patients with T2 diabetes and with or without CKD and controls without diabetes. They identified miR-190a-5p as decreased in the circulation of diabetic patients with CKD in comparison with diabetic patients without CKD. This miRNA has not been yet identified as a biomarker of renal function decline in CKD. Next, the authors wanted to test whether the expression of miR-190a-5p was correlated with renal function or renal function decline in a cohort of patients with CKD from different etiologies. Interestingly, absence of detectable expression of miR-190a-5p was associated with lower eGFR in the cohort. For patients with detectable miR-190a-5p expression, the authors found that expression below the median was associated with higher risk of renal function decline and ESKD in patients with no albuminuria or moderate albuminuria but not in patients with high albuminuria, suggesting 1) nephroprotective role of miR-190a-5p and 2) that miR-190a-5p could be used as a biomarker of CKD progression to ESKD in normo-albuminuric patients. Finally, the authors tried to link miR-190a-5p expression with renal pathophysiology. This part of the manuscript is the less convincing part and will require further investigation.

I have several suggestions to improve the quality of the manuscript:

- The link between low miR-190a-5p expression in the circulation and kidney function is not clear. The authors used a model of AKI (UUO model) to identify the cellular source of miR-190a-5p. Particularly they show strong decrease in its expression at the early stage of the model when the stage is still AKI and not yet CKD. The authors should verify that miR-190a-5p decreased expression in PT is found in other models of CKD such as DKD or glomerulopathies.
- similarly, the authors focused on targets in PT in the AKI model, whereas miR-190a-5p could target cells other than PT (immune cells, fibroblasts etc...) and the targets may be different in other disease context.
- A key demonstration of miR-190a-5p protective effects in CKD would be to the effects of miR-190a-5p mimics injection or genetic miR-190a-5p overexpression in mouse models of CKD.
- Regarding the in vitro experiments, mimic experiment induces a more than 100000 fold increase in miR-190a-5p expression (support fig 4: baseline 0,003 to mimic 1100). This appears to be completely out of the normal range of expression. With such an increased expression we may expect off-targets and adverse effects? Further, ADAM10 relative expression is only decreased by 40% while in vivo a 3 fold decrease of miR-190a-5p expression is sufficient to induce a two-fold increase of ADAM10 expression. Should we expect that such an increase of miR-190a-5p expression would induce a complete loss of ADAM10?
- One may wonder which signal triggers miR-190a-5p decrease during disease?

Reviewer #2

(Remarks to the Author)

In this study, the authors tried to perform small RNA-Sequencing (sRNA-Seq) to detect miRs and explore their potential as

biomarkers in prediction the progression of CKD. They firstly performed small RNA-Sequencing from the plasma of patients of DKD and without CKD and non-diabetic controls with normal kidney function. They found MiR-190a-5p abundance was significantly lower in the circulation of T2D patients with reduced kidney function compared to those with normal kidney function. Then they measured miR-190a-5p in an unselected cohort of CKD patients with no or moderate albuminuria (<300mg/mmol), and found serum miR-190a-5p levels predicted CKD progression. They further utilized transcriptomic data from mouse models of kidney injury and single nuclear (sn) RNA-Seq from human kidney to identify the kidney source of miR-190a-5p, and finding that miR-190a-5p is enriched in the proximal tubule (PT) but downregulated following injury. Bioinformatic analysis highlighted ADAM10 as a potential miR-190a5p target and we validated this in human PT cell line.

This is a descriptive study with no convincing conceptive merit. Many obvious shortcoming existed in the design.

1. The finding of the difference of MiR-190a-5p abundance is based on very small clinical samples. The evaluation of the progression of CKD is only based on the extent of albuminuria and eGFR, and there is no more convincing data such as renal pathological data.
2. DKD and CKD actually are the different etiological clinical diseases with different pathological features although there are some similar clinical features. Clearly the authors cannot use the data from DKD to predict CKD.
3. The bioinformatics data have not been confirmed in the human renal tissues which caused the lower credit of these findings as biomarker.

Reviewer #3

(Remarks to the Author)

The authors have undertaken an interesting investigation into profiling plasma miRs in patients with T2DM, both with and without kidney disease. They have extended miR-190a-5p, a candidate from the profiling, as a potential prognostic biomarker for CKD using the seNSOR study cohort. Next, they attempted to elucidate the pathophysiological role of renal miR-190a-5p using animal and cell culture models. The study's strength lies in its valid group classification, justified study inclusion/exclusion criteria, and statistical analysis for biomarker survey. The identified marker may diagnose CKD in patients with a no-low albuminuria category, addressing a current diagnostic gap. However, the study is preliminary and largely observational, lacking orthogonal validation even for a biomarker aspect and a clear attempt to determine the role of miR-190a-5p in CKD. This reviewer feels this manuscript does not provide any role of miR-190a-5p in kidney disease and any direct evidence supporting its marker validation, blood miR-190a-5p source from organ system, renal localization, and target gene regulation. The manuscript is not ready for publication and may be more suitable for a specialized kidney journal rather than a general medical science readership.

In the abstract, the statement "single nuclear RNA-seq from human kidney, finding that..." is misleading. There is currently no small RNA readout platform in sc/snRNA-seq approaches. The authors' assumption of TLN2 abundance as a surrogate, based on the intragenic locus of miR-190a-5p, is risky. The authors need to provide direct evidence of its Proximal Tubule (PT) expression. While the K-e-m from the previous JCI insight paper is much appreciated, the detected expression levels are not convincing. Furthermore, the validity of using miR expression data from the R-UUO model is highly questionable.

Figure 5 should be moved to Supplementary data to support the potential role of miR-190a-5p. All data presented are from another study. The authors need to validate ADAM10 as a target for miR-190a-5p rather than cherry-picking the target. And even there is no evidence in this study that miR-190a-5p targets ADAM10. What is the role of this interaction in CKD? In vitro miR-190a-5p inhibitor (~0.6-fold Dn)/mimic (~1000-fold Up) study could result in artifacts based on both non-canonical and canonical base pairing. The physiological role of the miR under CKD remains unclear.

On Page 3, the third paragraph should reflect the current state of miR-based biomarker development, considering the numerous studies conducted using NGS- or array-based miR approaches, even for kidney diseases. It should also be noted that the authors did not directly survey the CKD cohort with high-throughput approaches but instead used plasma from a diabetes cohort.

On Page 4, the last paragraph should provide FCs of other mentioned miRs. It might be more beneficial for biomarker assay development if the two miRs are not detected in a group with a low FDR.

In Supplementary Figure 1A, the rationale for performing PCA with the top 100 miRs is unclear. The meaning of "top 100 miRs" and the conclusion drawn from this analysis should be clarified. In Supplementary Figure 1B, it should be specified whether the counts are normalized and what the adjusted P-values between groups are.

Version 1:

Reviewer comments:

Reviewer #1

(Remarks to the Author)

The authors have considerably improved the article and have taken all my comments into consideration. In particular, they have added an in vivo experiment in mice, which supports the role of microRNA. I have no more questions or comments to add.

Reviewer #2

(Remarks to the Author)

Overall, the author addressed some of the concerns. However, the data regarding the expression and function of miR-190a-5p/ADAM10 in renal fibrosis in vivo is not sufficient and convincing. Also the conditional manipulation of those two targets genes in vivo is necessary .

Reviewer #3

(Remarks to the Author)

The authors have strengthened the manuscript by incorporating data from two additional models, subtotal Nx and uIRI, to support the biomarker potential of miR-190a-5p in kidney injury. These findings, along with the quantification of miR-190a-5p in renal biopsies and its correlation with tubular cell mass, provide compelling evidence for its relevance in CKD. Furthermore, the observed reduction in tubular injury and fibrosis following administration of the miR-190a-5p mimic in animal models adds potential therapeutic significance to the study.

Nevertheless, concerns remain regarding the proposed miR-190a-5p–ADAM10 axis as the underlying mechanism of protection. Although the authors present in vivo data showing reduced ADAM10 expression following miR-190a-5p mimic treatment—an important advance—these findings do not establish targeting of ADAM10 by miR-190a-5p, nor do they demonstrate that the observed reductions in injury and fibrosis are specifically mediated through this axis. Given the pleiotropic nature of miRNA regulation, the potential for non-canonical or indirect effects complicates the mechanistic interpretation. Without further experimental validation, it remains uncertain whether the protective effects are primarily driven by the miR-190a-5p–ADAM10 pathway.

We thank the Reviewers for their helpful comments on our manuscript “**Low circulating miR-190a-5p predicts progression of chronic kidney disease**”.

We appreciate that the reviewers found the manuscript interesting, and its strength lay in its valid group classification, justified study inclusion/exclusion criteria, and statistical analysis for biomarker survey. We believe that the revised manuscript has benefitted from the comments as our new data more robustly supports our original conclusions that miR-190a-5p is a prognostic biomarker, and additionally we now demonstrate that it may be a therapy to prevent AKI to CKD transition.

We have addressed each point individually:

Reviewer 1:

The link between low mir-190a-5p expression in the circulation and kidney function is not clear. The authors used a model of AKI (UUO model) to identify the cellular source of mir-190a-5p. Particularly they show strong decrease in its expression at the early stage of the model when the stage is still AKI and not yet CKD. The authors should verify that mir-190a-5p decreased expression in PT is found in other models of CKD such as DKD or glomerulopathies.

We provide additional data which demonstrates that circulating miR-190a-5p abundance correlates with expression of miR-190a-5p in the renal cortex in patients from our seNSOR study who had undergone renal biopsy. Both circulating and renal miR-190a-5p levels correlate with renal function (eGFR at the time of biopsy). These data are shown in new Figures 3C and D.

We also provide data that demonstrate reduced miR-190a-5p expression in the renal cortex in two additional pre-clinical models of kidney injury, the subtotal nephrectomy model of CKD, which exhibits glomerular injury, tubular atrophy, and fibrosis and the unilateral IRI, which models AKI-to-CKD transition. These data are shown in new Figures 4A and B.

Similarly, the authors focused on targets in PT in the AKI model, whereas mir-190a-5p could targets cells others than PT (immune cells, fibroblasts etc...) and the targets may be different in other disease context.

In the murine model of reversible ureteric obstruction, we had previously isolated key cell types by FACS and determined that miR-190a-5p was expressed at much higher levels in proximal tubular cells than in other cell types, such as immune cells or fibroblasts (Figure 4C).

As miRs are not captured by snRNA-seq, in our human studies we used expression of its host gene, *TLN2*, as a surrogate for miR-190a-5p. miR-190a-5p and *TLN2* share the same promoter and we observed a very tight correlation between miR-190a-5p and *Tln2* expression in murine kidneys. We determined that *TLN2* was expressed predominantly by tubular cells and its expression was reduced as tubules transition from health to an injured and inflamed state (Figure 6D). Hence, in both humans and animal models, our data suggest that tubular cells comprise the majority of miR-190a-5p expression. As miR abundance correlates with its efficacy in reducing target gene expression, we consider that miR-190a-5p is more likely to exert functional influence on tubular cells, although we cannot exclude a small effect in other cell types. Hence, we have included this statement in the manuscript: “miR-190a-5p was also expressed at a much lower level in other kidney cells, including macrophages, fibroblasts and endothelial cells, therefore we could not exclude that it may regulate gene expression in a more subtle manner in these cell types.

A key demonstration of mir-190a-5p protective effects in CKD would be to the effects of mir-190a-5p mimics injection or genetic mir-190a-5p overexpression in mouse models of CKD.

We have performed an additional experiment in which we administered a miR-190a-5p mimic or mimic control to mice undergoing unilateral ischaemia-reperfusion injury, a model of AKI-to-CKD transition. Administration of the mimic restored the reduction in miR-190a-5p expression observed in the renal cortex following IRI and this was associated with a reduction in tubular injury (*Havcr1* expression) and preserved expression of proximal tubular cell genes and a reduction in tubulointerstitial fibrosis. This data is presented as the new Figure 5.

Regarding the in vitro experiments, mimic experiment induces a more than 10000 fold increase in mir-190-5p expression (support fig 4: baseline 0,003 to mimic 1100). This appears to be completely out of the normal range of expression. With such an increased expression we may expect off-targets and adverse effects? Further, ADAM10 relative expression is only decreased by 40% while in vivo a 3 fold decrease of mir-190a-5p expression is sufficient to induce a two-fold increase of ADAM10 expression. Should we expect that such an increase of mir-190a-5p expression would induce a complete loss of ADAM10?

We have now performed dose-response studies using miR-190a-5p mimic in RPTECs, generating miR-150a-5p levels ranging from 300 to ~1500-fold greater than with control mimic, and observing a dose-dependent decrease in *ADAM10* expression. This is now included in an expanded Supplementary Figure 4. The non-linearity of these results with the mimic may be due to the complex nature of the biomachinery of miRNA repression, for example the loading onto other Ago proteins which do not reduce mRNA levels.

One may wonder which signal triggers mir-190a-5p decrease during disease?

We agree this is an interesting question. The literature suggests that miR-190a-5p overexpression can block TGF β -induced EMT. As TGF β is one of the best characterised pro-injury/pro-fibrotic cytokines in kidney injury we examined whether it altered miR-190a-5p expression in human renal proximal tubular cells and found that it had no effect on miR-190a-5p expression. We include this data here in the Response to Reviewers (Figure 1). Identifying the mechanism(s) that lead to reduced miR-190a-5p requires further investigation, which we consider outside the scope of this manuscript.

Figure 1: Stimulation of RPTECS with rhTGFb1 has no effect on miR-190a-5p expression. Human renal proximal tubular cells (RPTECs) were cultured in 6 well plates, placed into serum free media for 24hrs before stimulation with 10ng/mL rhTGFb1 for 72hrs. Cells were lysed with qiazol and RNA extracted. miR-190a-5p expression was determined using specific primer/probes and expression normalised to U6. n=3 biological replicates.

Reviewer 2:

This is a descriptive study with no convincing conceptive merit. Many obvious shortcomings existed in the design.

We have now performed additional experiments to address the shortcomings identified by the reviewer and importantly we have now employed a miR-190a-5p mimic in a pre-clinical model, which demonstrates a functional role for miR-190a-5p in maintaining tubular cell health after an acute injury.

1. The finding of the difference of MiR-190a-5p abundance is based on very small clinical samples. The evaluation of the progression of CKD is only based on the extent of albuminuria and eGFR, and there is no more convincing data such as renal pathological data.

Although the discovery sRNA-seq screen included a small number of patients, we determined that a low circulating miR-190a-5p level predicted hard clinical outcomes including the requirement for dialysis or a >30% decline in kidney function in a cohort of 298 patients with CKD of unselected aetiology. These are standard outcome measures that are employed in biomarker studies and clinical trials. To enable correlation of miR-190a-5p expression with important renal pathological data, we have additionally quantified miR-190a-5p expression in renal biopsy samples from patients with CKD. Renal miR-190a-5p expression correlated with tubular cell mass as determined by expression of the tubular epithelial markers CD10 and PanCK and was inversely correlated with tubulointerstitial fibrosis (new Figures 3C-F).

2. DKD and CKD actually are the different etiological clinical diseases with different pathological features although there are some similar clinical features. Clearly the authors cannot use the data from DKD to predict CKD.

Our sRNA-seq screen was designed primarily to identify miRs that were differentially abundant in the circulation in patients with reduced function compared with normal kidney function rather than those that were differentially expressed in those with diabetes versus non-diabetics. Indeed, we observed that a reduction in miR-190a-5p in the circulation reflected impaired renal function, rather than the diabetic state *per se*. We next wished to determine whether this was a feature specific to diabetic kidney disease or a generic response to kidney injury, therefore we assessed miR-190a-5p abundance in the circulation of patients with CKD of diverse aetiologies, including patients with diabetic nephropathy. In this cohort, miR-190a-5p abundance correlated with eGFR and a circulating miR-190a-5p level below the median predicted adverse renal outcomes.

3. The bioinformatics data have not been confirmed in the human renal tissues which caused the lower credit of these findings as biomarker.

We have now performed additional studies to quantify miR-190a-5p expression in frozen biopsy cores from patients who were recruited to our prospective study and underwent renal biopsy at the time of

recruitment. This confirmed that renal miR-190a-5p expression correlated with the circulating miR-190a-5p abundance and with renal function and tubular epithelial cell mass. This suggests that miR-190a-5p may be a biomarker of healthy tubular cell mass, which may be the reason why low miR-190a-5p levels predict adverse renal outcomes.

Reviewer 3:

The authors have undertaken an interesting investigation into profiling plasma miRs in patients with T2DM, both with and without kidney disease. They have extended miR-190a-5p, a candidate from the profiling, as a potential prognostic biomarker for CKD using the seNSOR study cohort. Next, they attempted to elucidate the pathophysiological role of renal miR-190a-5p using animal and cell culture models. The study's strength lies in its valid group classification, justified study inclusion/exclusion criteria, and statistical analysis for biomarker survey. The identified marker may diagnose CKD in patients with a no-low albuminuria category, addressing a current diagnostic gap. However, the study is preliminary and largely observational, lacking orthogonal validation even for a biomarker aspect and a clear attempt to determine the role of miR-190a-5p in CKD. This reviewer feels this manuscript does not provide any role of miR-190a-5p in kidney disease and any direct evidence supporting its marker validation, blood miR-190a-5p source from organ system, renal localization, and target gene regulation. The manuscript is not ready for publication and may be more suitable for a specialized kidney journal rather than a general medical science readership.

We are grateful that the reviewer acknowledged the strengths of the previous study including the validity of the recruitment and analysis and finding that miR-190a-5p predicts renal outcomes in patients with no-low albuminuria meets a current diagnostic gap.

While our previous manuscript was largely observational in nature, we now provide additional data to support a biological role for miR-190a-5p in the pathogenesis of kidney disease. Mitigating loss of miR-190a-5p in the kidney following ischaemic injury by administration of a miR-190a-5p mimic reduced expression of markers of tubular cell injury, preserved expression of markers of healthy tubular cells and reduced renal fibrosis (new Figure 5). These exciting new data suggest that, in addition to being a biomarker of renal prognosis, miR-190a-5p may represent an attractive therapeutic target to prevent AKI-to-CKD transition.

In the abstract, the statement "single nuclear RNA-seq from human kidney, finding that..." is misleading. There is currently no small RNA readout platform in sc/snRNA-seq approaches. The authors' assumption of TLN2 abundance as a surrogate, based on the intragenic locus of miR-190a-5p, is risky. The authors need to provide direct evidence of its Proximal Tubule (PT) expression. While the K-e-m from the previous JCI insight paper is much appreciated, the detected expression levels are not convincing. Furthermore, the validity of using miR expression data from the R-UUO model is highly questionable.

In our pre-clinical model, we isolated key renal cell types by FACS, including proximal tubular cells, immune cells, endothelial cells and fibroblasts and we determined that miR-190a-5p was approximately 5-10-fold higher than in the other cell types (Figure 4C). We agree that current sc/snRNA-seq approaches cannot quantify miR expression, hence we employed its host gene *TLN2* as a surrogate of miR-190a-5p expression, given that they both utilise the same promoter and expression of *Tln2* and miR-190a-5p correlated tightly in our pre-clinical models (new Figure 4E). We have now

additionally quantified miR-190a-5p expression in frozen cores of renal biopsies from patients with CKD. The small amount of biopsy tissue available made it impossible to quantify miR-190a-5p specifically in the tubular compartment, however we did find that miR-190a-5p expression correlated closely with tubular cell mass as quantified by immunofluorescence staining with CD10 and PanCK and inversely with fibrosis. Taken together, these data support the statement that miR-190a-5p is predominantly expressed by tubular cells rather than fibroblasts or immune cells. However, we appreciate that further studies are required to confirm this. We have therefore included this statement in the discussion: 'miR-190a-5p was also expressed at a much lower level in other kidney cells, including macrophages, fibroblasts and endothelial cells, therefore we could not exclude that it may regulate gene expression in a more subtle manner in these cell types'.

Figure 5 should be moved to Supplementary data to support the potential role of miR-190a-5p. All data presented are from another study. The authors need to validate ADAM10 as a target for miR-190a-5p rather than cherry-picking the target. And even there is no evidence in this study that miR-190a-5p targets ADAM10. What is the role of this interaction in CKD? In vitro miR-190a-5p inhibitor (~0.6-fold Dn)/mimic (~1000-fold Up) study could result in artifacts based on both non-canonical and canonical base pairing. The physiological role of the miR under CKD remains unclear.

We have supplemented the data from the original study in the rUUO model with new data from 2 additional pre-clinical models, subtotal nephrectomy and ischaemia-reperfusion (new Figure 4). These data were not included in our previous paper, rather the raw dataset was published and we performed new analysis of the data pertaining to miR-190a-5p specifically. Pending the thoughts of the reviewer and editor, we are happy to consider moving new Figure 4 to the supplementary information, however we do consider that the data are important for the context of the manuscript.

To mitigate the possibility that the down-regulation of *ADAM10* observed with miR-190a-5p was due to non-canonical base pairing, we compared the miR-190a-5p mimic with a negative control mimic at the same dose (Figure 6E). Importantly, our new data show that the miR-190a-5p mimic also reduced *Adam10* expression in the kidney in vivo (Figure 6F). In this study the renal miR-190a-5p expression was ~2-fold higher with miR-190a-5p mimic than in control mimic and similar to levels observed in sham-operated kidneys (Figure 4A and B), providing further support that miR-190a-5p can module *ADAM10* at pathophysiological levels of renal expression in the context of a pre-clinical model of AKI-to-CKD transition. We cannot exclude the possibility that some of the effect of miR-190a-5p on *ADAM10* may be indirect, therefore we have expanded on this in our discussion, highlighting that **“Indeed, our pre-clinical data suggests that miR-190a-5p may maintain kidney cell health in part by inhibiting *ADAM10* expression either directly or indirectly as *Adam10* knockout protects against kidney injury (41).”**

On Page 3, the third paragraph should reflect the current state of miR-based biomarker development, considering the numerous studies conducted using NGS- or array-based miR approaches, even for kidney diseases. It should also be noted that the authors did not directly survey the CKD cohort with high-throughput approaches but instead used plasma from a diabetes cohort.

We have now altered the text to reference studies that have used NGS or array-based miR discovery approaches. The text now reads “The development of high-throughput next-generation sequencing (NGS) technologies offers new opportunities for unbiased quantification of miRs (28, 29) and has led

to the identification of potential miRNA-based biomarkers, including for paediatric transplant rejection, CKD and IgA Nephropathy (30, 31, 32).”

Our initial sRNA-seq screen was designed primarily to identify miRs that were differentially abundant in the circulation in patients with reduced function compared with normal kidney function rather than those that were differentially expressed in those with diabetes versus non-diabetics. Indeed, we observed that a reduction in miR-190a-5p in the circulation reflected impaired renal function, rather than the diabetic state *per se*, which led us to assess the role of miR-190a-5p in an unselected cohort of patients with CKD.

On Page 4, the last paragraph should provide FCs of other mentioned miRs. It might be more beneficial for biomarker assay development if the two miRs are not detected in a group with a low FDR.

The logFC values have now been added to the manuscript for both miRs.

In Supplementary Figure 1A, the rationale for performing PCA with the top 100 miRs is unclear. The meaning of "top 100 miRs" and the conclusion drawn from this analysis should be clarified. In Supplementary Figure 1B, it should be specified whether the counts are normalized and what the adjusted P-values between groups are.

We apologise for our lack of clarity. We performed the PCA on the top 100 most differentially expressed miRNAs (normalised counts) across the groups. Differential expression analysis was performed on normalized counts principally using EdgeR.

For Supplementary 1A and 1B we have edited the figure legends for clarity.

‘Supplementary Figure 1: Small RNA sequencing of the circulating miRNome .

A. Principal component analysis (PCA) of the top 100 most variant miRs (normalised counts) across NDNRF, T2DNRF and T2DKD samples B. Expression of miR-190a-5p normalized counts (CPM) in the discovery patient cohort. NDNRF – Non-diabetic controls with renal function, T2DNF – Type 2 Diabetes with normal renal function, T2DKD - Type 2 Diabetes with reduced renal function.’

We thank the Reviewers for their helpful comments on our manuscript “**Low circulating miR-190a-5p predicts progression of chronic kidney disease**”.

We are grateful that the Editorial decision was to accept in principle.

We have addressed the remaining reviewer comments individually:

Reviewer #1 (Remarks to the Author):

The authors have considerably improved the article and have taken all my comments into consideration. In particular, they have added an in vivo experiment in mice, which supports the role of microRNA. I have no more questions or comments to add.

We thank the Reviewer for their positive comments on our manuscript.

Reviewer #2 (Remarks to the Author):

Overall, the author addressed some of the concerns. However, the data regarding the expression and function of miR-190a-5p/ADAM10 in renal fibrosis in vivo is not sufficient and convincing. Also the conditional manipulation of those two targets genes in vivo is necessary.

We agree that further follow-up experiments would be necessary to fully demonstrate the miR-190a-p/ADAM-10 axis. Therefore, we have added the following the results section:

“Hence, we confirm that miR-190a-5p may inhibit ADAM10 expression in tubular cells, however we cannot define if this is an indirect or direct effect on ADAM10.”

To the discussion we have added the following to further address the Reviewer’s comments:

“We demonstrate that Adam10 expression changes reciprocally to that of miR-190a-5p in the pre-clinical model of uIRI, with supplemented miR-190a-5p expression profoundly reducing Adam10 expression in vivo and in vitro, suggesting that ADAM10 may be a direct or indirect target of miR-190a-5p. From our data we postulate that the loss of miR-190a-5p in kidney disease may lead to aberrant expression of ADAM10 but further study would be required to provide additional evidence to support this. Further experiments would be required to demonstrate if ADAM10 is a direct target of miR-190a-5p, such as employing a 3’UTR binding assay. Our pre-clinical data suggests that miR-190a-5p may maintain kidney cell health in part by inhibiting ADAM10 expression, and this fits with other data which has demonstrated that Adam10 knockout protects against kidney injury (41). However, this requires confirmation in additional studies such as by demonstrating that the protective effect of the miR-190a-5p mimetic is abrogated in Adam10 knockout mice, and that conditional tubular knockout of miR-190a exacerbates kidney injury through elevated ADAM10 expression.”

Reviewer #3 (Remarks to the Author):

The authors have strengthened the manuscript by incorporating data from two additional models, subtotal Nx and uIRI, to support the biomarker potential of miR-190a-5p in kidney injury. These

findings, along with the quantification of miR-190a-5p in renal biopsies and its correlation with tubular cell mass, provide compelling evidence for its relevance in CKD. Furthermore, the observed reduction in tubular injury and fibrosis following administration of the miR-190a-5p mimic in animal models adds potential therapeutic significance to the study.

Nevertheless, concerns remain regarding the proposed miR-190a-5p–ADAM10 axis as the underlying mechanism of protection. Although the authors present *in vivo* data showing reduced ADAM10 expression following miR-190a-5p mimic treatment—an important advance—these findings do not establish targeting of ADAM10 by miR-190a-5p, nor do they demonstrate that the observed reductions in injury and fibrosis are specifically mediated through this axis. Given the pleiotropic nature of miRNA regulation, the potential for non-canonical or indirect effects complicates the mechanistic interpretation. Without further experimental validation, it remains uncertain whether the protective effects are primarily driven by the miR-190a-5p–ADAM10 pathway.

We believe we have addressed this Reviewer's comments in our response to Reviewer 2. These include detailing in the Discussion of the manuscript the further experiments that would be required to fully demonstrate that the protective effects are primarily driven by the miR-190a-5p–ADAM10 pathway.

“From our data we postulate that the loss of miR-190a-5p in kidney disease may lead to aberrant expression of ADAM10 but further study would be required to provide additional evidence to support this. Further experiments would be required to demonstrate if ADAM10 is a direct target of miR-190a-5p, such as employing a 3'UTR binding assay. Our pre-clinical data suggests that miR-190a-5p may maintain kidney cell health in part by inhibiting ADAM10 expression, and this fits with other data which has demonstrated that Adam10 knockout protects against kidney injury (41). However, this requires confirmation in additional studies such as by demonstrating that the protective effect of the miR-190a-5p mimetic is abrogated in Adam10 knockout mice, and that conditional tubular knockout of miR-190a exacerbates kidney injury through elevated ADAM10 expression.”